# *Unlearning Isn't Deletion*: INVESTIGATING REVERSIBILITY OF MACHINE UNLEARNING IN LLMs

## ABSTRACT

Unlearning in large language models (LLMs) aims to remove specified data, but its efficacy is typically assessed with task-level metrics like accuracy and perplexity. We demonstrate that these metrics are often misleading, as models can appear to forget while their original behavior is easily restored through minimal fine-tuning. This phenomenon of *reversibility* suggests that information is merely suppressed, not genuinely erased. To address this critical evaluation gap, we introduce a *representation-level analysis framework*. Our toolkit comprises PCA-based similarity and shift, centered kernel alignment (CKA), and Fisher information, complemented by a summary metric, the mean PCA distance, to measure representational drift. Applying this framework across six unlearning methods, three data domains, and two LLMs, we identify four distinct forgetting regimes based on their *reversibility* and *catastrophicity*. Our analysis reveals that achieving the ideal state–irreversible, non-catastrophic forgetting–is exceptionally challenging. By probing the limits of unlearning, we identify a case of seemingly irreversible, targeted forgetting, offering new insights for designing more robust erasure algorithms. Our findings expose a fundamental gap in current evaluation practices and establish a representation-level foundation for trustworthy unlearning.[1]

## 1 INTRODUCTION

Large language models (LLMs), trained on massive corpora, have achieved remarkable success across diverse tasks, yet their capacity to memorize training snippets poses acute ethical, legal, and security risks. Memorization can unintentionally disclose sensitive, harmful, or copyrighted text [30; 15; 39], conflicting with emerging regulations, such as the EU's *Right to be Forgotten* [9].

*Machine unlearning* seeks to address this challenge by algorithmically erasing the influence of specified data, making a model behave as if it had never been trained on that data [2]. While numerous unlearning methods have been developed for LLMs [44; 13; 32; 21; 20; 23; 41], their efficacy is typically assessed using task-level metrics, such as accuracy on a held-out "forget set."

However, these evaluations overlook a pivotal question: ***Does LLM unlearning achieve genuine erasure, or merely suppress information that can resurface?*** If supposedly erased knowledge is readily revived, unlearning constitutes a shallow perturbation with limited safety.

Emerging evidence indicates that many unlearning methods are superficially effective. After unlearning, models often show degraded performance on the forget set; yet, the "forgotten" knowledge can be rapidly recovered through minimal fine-tuning even on unrelated data [26; 28] (see Figure 1), low-bit quantization [48], or adversarial prompting [31; 25]. Although previous studies have identified this *reversibility* and the risks of catastrophic forgetting under accumulated updates (of repeated requests) [36], the representational dynamics governing these regimes have yet to be investigated.

This paper presents the **first systematic, representation-level analysis of LLM unlearning reversibility**. We demonstrate that task-level metrics (*e.g.*, forget accuracy) are insufficient to distinguish reversible forgetting from catastrophic failure, as surface-level performance collapse may occur while internal representations remain intact. To move beyond surface effects, a unified diagnostic toolkit: PCA subspace similarity and shift [49], centered kernel alignment (CKA) [17], and Fisher information

---

[1] https://anonymous.4open.science/r/Feature_tools_unlearning-BACA/

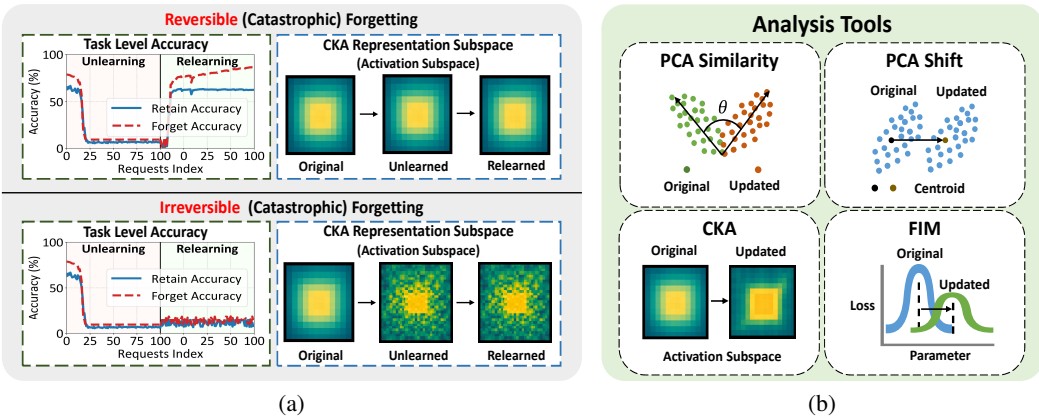

Figure 1: (a) task-level accuracy and CKA subspaces of **reversible** (top) vs. **irreversible** (bottom) catastrophic forgetting due to *continual unlearning* then *relearning*, (b) Our four diagnostic tools

(FIM) [3]. We further propose the *mean PCA distance* as a quantitative measure of representational drift, helping reveal how different unlearning regimes emerge.

With our toolkit, we build a taxonomy characterizing unlearning along two axes: reversibility and catastrophicity (collateral damage to retained knowledge). This allows us to distinguish four regimes:
1) *Reversible, Catastrophic:* Global performance collapse that is fully recoverable via relearning.
2) *Reversible, Non-Catastrophic:* Targeted performance modest degradation that is easily restored.
3) *Irreversible, Catastrophic:* Permanent and unrecoverable global performance collapse.
4) *Irreversible, Non-Catastrophic:* Ideally, permanent erasure of target data without collateral damage.

Crucially, alternative relearning strategies such as prompt attacks [31], jailbreaking [25], quantization [48], and in-context recovery (with five-shot demonstrations of the forget-set) "fail" once the model enters *reversible, catastrophic* forgetting. Since these methods involve minimal or no parameter updates (on unlearned models), they cannot restore the lost representations. Consequently, we employ relearning as a robust probe to investigate unlearning behavior. This approach allows us to unify single and continual unlearning under a single taxonomy, elucidating how distinct forgetting regimes emerge from request volume, hyperparameters, and the unlearning method itself. By further analyzing sample efficiency across data types, we conclude that genuine unlearning demands irreversible, non-catastrophic erasure rather than superficial degradation in task-level metrics.

**Contributions.** We summarize our main contributions as follows:

- We present the *first* systematic study of *reversibility* in both *single* and *continual* LLM unlearning. We introduce a representation-level diagnostic toolkit and a quantitative metric, the *mean PCA distance*, to analyze representational drift and distinguish four regimes of forgetting.

- We conduct extensive experiments with six unlearning methods on multiple LLMs across three distinct domains. Our results demonstrate that standard task-level metrics (*e.g.*, accuracy, perplexity, MIA susceptibility) are insufficient for assessing the true extent of unlearning. and we further find that relearning exhibits different sample efficiencies depending on the type of data.

- We theoretically analyze weight perturbations to explain how widespread vs. localized parameter changes relate to (ir)reversible forgetting. Small perturbations near the logits can distort task-level metrics despite intact features, hence leading to misleading assessments.

- We identify a case of *seemingly irreversible, non-catastrophic forgetting*, offering insights for designing more robust unlearning algorithms. We also highlight the potential for unlearning to serve as a form of data augmentation, improving model representations upon relearning.

## 2 PRELIMINARIES AND OUR FORMULATION

**LLM unlearning** aims to remove the influence of specific data from a trained model to enhance privacy, improve safety, or mitigate bias [44; 13; 32; 21; 20; 23]. The standard paradigm involves a

Table 1: Four regimes of forgetting characterized by the *reversibility* and *catastrophicity*: ● denotes regimes commonly observed in practice, and ◑ denotes the ideal but elusive regime.

| Regime | Observed | Description |
|---|---|---|
| Reversible, Catastrophic | ● | Performance on both forget and retain sets collapses, but is recoverable via relearning. |
| Reversible, Non-Catastrophic | ● | Targeted performance drops on the forget set, which can be easily restored. |
| Irreversible, Catastrophic | ● | Global, unrecoverable performance collapse on both forget and retain sets. |
| Irreversible, Non-Catastrophic | ◑ | Targeted, permanent erasure of forget-set knowledge with no collateral damage. |

training corpus $\mathcal{D}$, from which $\mathcal{D}_f \subseteq \mathcal{D}$ is designated as the *forget set*. A model $\mathcal{M}$ is first trained on $\mathcal{D}$ via an algorithm $\mathcal{A}$. An unlearning procedure $\mathcal{U}$ then transforms $\mathcal{M}$ into *unlearned* $\mathcal{M}_f$, which should ideally behave as if it were trained only on the *retain set* $\mathcal{D}_r = \mathcal{D} \setminus \mathcal{D}_f$. Formally, the goal is to statistically approximate the retrained model $\mathcal{M}_r$: $\mathcal{M}_f = \mathcal{U}(\mathcal{M}, \mathcal{D}_f) \approx \mathcal{M}_r = \mathcal{A}(\mathcal{D}_r)$.

Retraining LLMs is prohibitively costly, so most studies rely on empirical proxies rather than formal statistically-indistinguishable guarantees [29; 21; 7]. Evaluations track *forget quality* on the forget set, *utility*, and downstream task *accuracy* on the retain set, aiming to preserve both dimensions.

While current methods can achieve reasonable balances between forgetting and utility in *single-shot* scenarios [1; 8], they often falter in the practical *continual* setting, where removal requests arrive sequentially over time [1]. For a sequence of forget sets $\mathcal{D}_f^{(1)}, \mathcal{D}_f^{(2)}, \ldots, \mathcal{D}_f^{(t)}$ (the union is $\mathcal{D}_f$), the retain set is $\mathcal{D}_r^{(t)}$ after $t$ rounds. The model is then updated recursively: $\mathcal{M}_f^{(t)} = \mathcal{U}(\mathcal{M}_f^{(t-1)}, \mathcal{D}_f^{(t)})$, which should be similar to $\mathcal{M}_r = \mathcal{A}(\mathcal{M}, \mathcal{D}_r^{(t)}), \forall t$. However, empirically, it often leads to *catastrophic forgetting*–a severe decline in performance on both forgotten and retained knowledge [1; 36].

Single-shot unlearning is "fragile:" fine-tuning, even on benign, unrelated data, can rapidly restore the supposedly "forgotten" knowledge [1; 28; 26]. Such fragility persists in *continual* unlearning as well. Prior work has noted this phenomenon but has not deeply investigated its underlying mechanics.

## 2.1 A Taxonomy of Forgetting Regimes

We hypothesize that this performance collapse does not necessarily equate to true information erasure; the knowledge might merely become latent or suppressed. To formalize this hypothesis, we introduce a taxonomy of forgetting based on two axes: **catastrophicity** (the extent of collateral damage to retained knowledge) and **reversibility** (whether forgotten knowledge can be recovered).

Let $\theta_0$ be the initial model parameters, $\theta_u$ be the parameters after unlearning, and $\theta_r$ be the parameters after a subsequent *relearning* phase (defined below). We use $E(\theta, \mathcal{T})$ to denote a performance metric (*e.g.*, accuracy) evaluated on a task set $\mathcal{T}$, which can be partitioned into a forget-related task $\mathcal{T}_f$ and a retain-related task $\mathcal{T}_r$. We define four distinct regimes of forgetting, summarized in Table 1.

**Definition 1** (**Four Regimes of Forgetting**). Let $\Delta_u(\mathcal{T}) = E(\theta_0, \mathcal{T}) - E(\theta_u, \mathcal{T})$ be the performance drop after unlearning, and $\Delta_r(\mathcal{T}) = E(\theta_0, \mathcal{T}) - E(\theta_r, \mathcal{T})$ be the residual drop after relearning. The nature of forgetting is determined by these drops on the forget set ($\mathcal{T}_f$) and retain set ($\mathcal{T}_r$).

**Catastrophic** vs. **Non-Catastrophic**: Forgetting is *catastrophic* if utility on the retain set degrades significantly (both $\Delta_u(\mathcal{T}_r) \text{ and } \Delta_u(\mathcal{T}_u) \gg 0$) and *non-catastrophic* otherwise ($\Delta_u(\mathcal{T}_r) \approx 0$).

**Reversible** vs. **Irreversible**: Forgetting is *reversible* if relearning almost recovers initial performance on forget set ($\Delta_r(\mathcal{T}_f) \approx 0$) and *irreversible* if a significant performance drop persists ($\Delta_r(\mathcal{T}_f) \gg 0$).

The combination of these two properties yields four regimes, among which the *irreversible, non-catastrophic* forgetting is deemed ideal, but remains challenging to achieve in practice.

*Relearning Restriction.* Comparative analysis (see Appendix A.4.1) reveals that only relearning attacks reliably restore forgotten knowledge; we therefore employ relearning as our primary empirical probe to investigate forgetting regimes. To rigorously test the reversibility, we define a constrained relearning protocol that is distinct from full retraining. Given an unlearned model parameterized by $\theta_u$, we obtain the recovered model $\theta_r$ via brief fine-tuning on a restricted dataset, without access to the raw pre-training corpus. The relearning budget is strictly matched to the forget set size ($|\mathcal{D}_f|$), with data drawn from one of three sources: (i) the forget set $\mathcal{D}_f$ itself (representing a worst-case recovery scenario), (ii) a domain-aligned retain subset $\mathcal{D}_r^{(t)}$, or (iii) general out-of-distribution (or unrelated) data. We further analyze how sample efficiency varies across these data types in Appendix A.4.2.

Table 2: Yi-6B: MIA / F.Acc / R.Acc (%) simple task using three LRs under single unlearning, relearn by fine-tuning once on the entire forget set

| Phase | Method | LR=$3\times10^{-6}$ | | | LR=$4\times10^{-6}$ | | | LR=$5\times10^{-6}$ | | |
|---|---|---|---|---|---|---|---|---|---|---|
| | | MIA | F.Acc | R.Acc | MIA | F.Acc | R.Acc | MIA | F.Acc | R.Acc |
| Original | – | 70.9 | 78.9 | 65.5 | 70.9 | 78.9 | 65.5 | 70.9 | 78.9 | 65.5 |
| Unlearn | GA | 45.5 | 65.4 | 54.0 | 43.8 | 62.4 | 52.3 | 41.2 | 60.3 | 50.9 |
| | GA+GD | 65.4 | 75.1 | 64.6 | 58.2 | 73.8 | 65.8 | 55.3 | 68.5 | 63.5 |
| | GA+KL | 48.9 | 71.0 | 58.5 | 47.6 | 70.6 | 58.1 | 44.8 | 68.4 | 55.4 |
| | NPO | 67.2 | 76.2 | 64.7 | 65.2 | 75.8 | 62.8 | 62.2 | 75.2 | 62.7 |
| | NPO+KL | 66.5 | 76.3 | 64.8 | 67.2 | 76.4 | 63.2 | 64.5 | 75.6 | 61.2 |
| | RLabel | 69.6 | 77.7 | 64.7 | 69.2 | 76.5 | 64.5 | 68.7 | 75.4 | 63.3 |
| Relearn | GA | 67.2 | 76.6 | 65.2 | 68.6 | 77.6 | 62.8 | 67.6 | 76.9 | 65.5 |
| | GA+GD | 68.6 | 77.0 | 65.3 | 68.8 | 76.9 | 65.3 | 68.8 | 77.2 | 65.3 |
| | GA+KL | 67.9 | 77.6 | 65.3 | 68.3 | 75.5 | 65.2 | 67.7 | 77.2 | 65.2 |
| | NPO | 68.2 | 77.1 | 65.3 | 68.2 | 77.2 | 65.2 | 68.3 | 77.0 | 65.1 |
| | NPO+KL | 68.9 | 77.1 | 65.3 | 67.9 | 76.3 | 63.0 | 68.6 | 76.9 | 65.2 |
| | RLabel | 68.3 | 78.8 | 65.6 | 68.9 | 76.4 | 65.3 | 68.8 | 78.9 | 65.2 |

## 3 CLASSIC (TASK-LEVEL) EVALUATION CAN BE DECEPTIVE

### 3.1 EXPERIMENT SETUP

**Models and Datasets.** We adopt two open-source LLMs: Yi-6B [45] and Qwen-2.5-7B [42]. To assess the generality of our findings, we employ two distinct dataset types for unlearning: (i) *simple tasks*, comprising arXiv abstracts and GitHub code from [44], and (ii) a *complex task*, NuminaMath-1.5, a recent benchmark for mathematical reasoning [19]. All experiments are performed on NVIDIA H100 GPUs. (Additional results on TOFU [29] and the Traditional-Chinese corpus[2] are in Appendix.)

**Unlearning algorithms.** We evaluate six canonical unlearning methods, organized into three families.

1) Gradient-Ascent (GA) family. The unified goal is $\mathcal{L} = \mathcal{L}_{\text{forget}}(\mathcal{D}_f) + \lambda\,\mathcal{L}_{\text{retain}}(\mathcal{D}_r)$, where $\mathcal{L}_{\text{forget}}$ maximizes the loss on the forget set via GA, $\mathcal{L}_{\text{retain}}$ (optional) preserves utility on the retain set, and $\lambda > 0$ balances the two. Choices for $\mathcal{L}_{\text{retain}}$ give three variants: i) GA ($\mathcal{L}_{\text{retain}} = 0$), ii) GA+GD (standard cross-entropy on $\mathcal{D}r$), and iii) GA+KL (KL divergence to the reference model on $\mathcal{D}r$) [44].

2) Negative Preference Optimization (NPO) family. GA is replaced by an NPO loss that penalizes agreement with the forget set [47]: $\mathcal{L} = \mathcal{L}_{\text{NPO}}(\mathcal{D}_f) + \lambda\,\mathcal{L}_{\text{retain}}(\mathcal{D}_r)$, Variants mirror those above: NPO ($\mathcal{L}_{\text{retain}} = 0$) and NPO+KL (retain-set KL regularization).

3) Random Label (RLabel). To mimic a model that never saw $\mathcal{D}_f$, true labels are replaced with random ones: $\mathcal{L} = \mathcal{L}_{\text{RLabel}}(\mathcal{D}_f)$, inducing near-uniform predictions without GA/negative rewards [44].

**Unlearning Scenarios.** We consider two standard settings: i) **Single unlearning:** A trained model $\mathcal{M}$ receives exactly one request to remove $\mathcal{D}_f \subset \mathcal{D}$, and ii) **Continual unlearning:** The model processes a stream of requests $\mathcal{D}_f^{(1)}, \ldots, \mathcal{D}_f^{(t)}$, yielding a sequence of models where $\mathcal{M}^{(t)} = \mathcal{U}(\mathcal{M}^{(t-1)}, \mathcal{D}_f^{(t)})$.

For *simple* tasks, we benchmark all six algorithms. For the *complex* math reasoning task, where a well-defined retain set is not available, we evaluate the core GA, NPO, and RLabel methods.

**Evaluation Metrics.** In *single* unlearning (simple tasks), we measure forget-set accuracy (F.Acc), retain-set accuracy (R.Acc), and privacy leakage via min-$k$%-prob MIA AUC [35].

In *continual* unlearning (both task types), we provide a more comprehensive evaluation. For simple tasks, we report: F.Acc / R.Acc, forget/retain perplexity (F.Ppl / R.Ppl), downstream accuracy on CommonsenseQA (CSQA )and GSM8K$_{\text{0-shot}}$ [37; 4], and min-$k$%-prob MIA AUC. For the complex task, we employ MATH$_{\text{0-shot}}$ [10] and GSM8K$_{\text{0-shot}}$ as the primary math utility benchmarks.

**Relearning Setting.** To assess the *reversibility* and of unlearning, each run is followed by a controlled *relearning* phase. The unlearned model is briefly fine-tuned on specific data without access to the full pre-training corpus. For **single unlearning**, we fine-tune once on the entire forget set $\mathcal{D}_f$.

---

[2]https://huggingface.co/datasets/taide/taide-bench

Table 3: Yi-6B: MIA / F.Acc / R.Acc (%) for simple task under four unlearning settings. Bold numbers indicate improvements over the Original baseline in F.Acc or R.Acc. The relearning phase uses the cumulative forget set.

| Phase | Method | LR=$3\times10^{-5}$, N=100 | | | LR=$5\times10^{-6}$, N=100 | | | LR=$3\times10^{-6}$, N=100 | | | LR=$3\times10^{-5}$, N=6 | | |
|-------|--------|------|-------|-------|------|-------|-------|------|-------|-------|------|-------|-------|
| | | MIA | F.Acc | R.Acc | MIA | F.Acc | R.Acc | MIA | F.Acc | R.Acc | MIA | F.Acc | R.Acc |
| Original | —— | 70.8 | 78.9 | 65.5 | 70.8 | 78.9 | 65.5 | 70.8 | 78.9 | 65.5 | 70.8 | 78.9 | 65.5 |
| Unlearn | GA | 26.1 | 0.0 | 0.0 | 23.2 | 9.1 | 6.2 | 25.2 | 16.8 | 14.4 | 29.6 | 36.3 | 36.1 |
| | GA+GD | 16.8 | 9.7 | 2.3 | 28.7 | 3.6 | 3.1 | 69.4 | 78.8 | 65.5 | 66.9 | 77.0 | 64.0 |
| | GA+KL | 17.8 | 9.0 | 6.2 | 27.3 | 9.1 | 6.2 | 18.9 | 3.8 | 3.2 | 29.5 | 52.9 | 41.5 |
| | NPO | 60.1 | 37.8 | 37.9 | 50.6 | 51.0 | 52.3 | 68.4 | 78.3 | 64.1 | 68.7 | 71.6 | 59.4 |
| | NPO+KL | 59.0 | 64.3 | 55.9 | 65.4 | 77.6 | 64.3 | 66.7 | 78.8 | 65.5 | 67.9 | 67.6 | 56.1 |
| | RLabel | 65.1 | 0.0 | 0.0 | 63.6 | 0.1 | 0.4 | 61.4 | 0.4 | 0.7 | 62.7 | 72.7 | 61.1 |
| Relearn | GA | 74.5 | 2.1 | 1.8 | 68.0 | **80.0** | 65.0 | 68.6 | **80.8** | 65.2 | 68.2 | 70.5 | 58.7 |
| | GA+GD | 68.1 | 2.2 | 2.6 | 69.8 | **81.2** | 65.1 | 70.0 | **81.8** | 65.5 | 67.0 | 61.6 | 54.4 |
| | GA+KL | 70.7 | 1.7 | 1.6 | 68.3 | **81.1** | 64.8 | 70.7 | **81.0** | 63.2 | 65.0 | 66.6 | 56.2 |
| | NPO | 70.0 | 57.0 | 45.6 | 68.0 | **82.7** | 65.5 | 69.9 | **81.2** | 65.5 | 68.4 | 71.2 | 59.4 |
| | NPO+KL | 67.7 | 60.7 | 54.2 | 69.5 | **83.8** | **65.6** | 69.9 | **83.8** | 65.5 | 69.0 | 67.6 | 56.1 |
| | RLabel | 69.5 | 4.3 | 2.8 | 70.4 | **80.8** | 65.3 | 70.0 | **80.5** | 65.3 | 65.2 | 72.7 | 61.1 |

For **continual unlearning**, we evaluate three conditions: (i) the cumulative forget set $\bigcup_t \mathcal{D}_f^{(t)}$, representing a worst-case adversarial scenario, (ii) the corresponding retain subset $\mathcal{D}_r^{(t)}$, as a proxy for the data distribution, and (iii) unrelated out-of-distribution data (general-domain samples explicitly different from $\mathcal{D}_f$). Each relearning dataset is size-matched to its corresponding unlearning request.

**Hyperparameter Configuration.** To comprehensively evaluate the effects of unlearning, we design multiple hyperparameter configurations that vary both the learning rate and the number of unlearning requests. For single unlearning we sweep the learning rate over LR $\in \{3, 4, 5\} \times 10^{-6}$ while fixing the request count to $N = 1$. For continual unlearning we vary both knobs: on the simple task (Yi-6B) we test LR $\in \{3, 5\} \times 10^{-6} \cup \{3 \times 10^{-5}\}$ with $N \in \{6 \to 100\}$; on the complex task (Qwen-2.5-7B) we use LR $\in \{3, 5\} \times 10^{-6}$ and $3 \times 10^{-5}$ together with $N \in \{6 \to 100\}$. All runs adopt the optimizer settings of [38]: AdamW [27] ($\beta_1 = 0.9$, $\beta_2 = 0.95$, $\varepsilon = 10^{-8}$), a cosine schedule with 10% warm-up followed by decay to 10% of peak, weight decay 0.1, and gradient clipping at 1.0.

## 3.2 EVALUATION RESULTS

We report quantitative results for single and continual unlearning on Yi-6B and Qwen-2.5-7B under various configurations. Complete results are provided in Appendix Tables 8 and 9.

**Single Unlearning.** On Yi-6B, all six methods successfully reduce MIA and F.Acc, indicating a certain degree of forgetting (Table 2). The impact on the retain set is modest, with R.Acc dropping by only 2–5%. However, relearning often restores original performance; for instance, GA+KL and RLabel recover R.Acc to approximately 65% and F.Acc above 77%. These findings suggest that single unlearning achieves superficial forgetting, as the underlying representations remain largely intact (Section 4.2.1). This outcome characterizes the *reversible, non-catastrophic forgetting* regime.

**Continual Unlearning.** Post-relearning analysis (Tables 3, 8, and 9) reveals two forms of reversible forgetting. In *reversible, catastrophic forgetting*, both utility (*e.g.*, F.Acc, R.Acc) and privacy metrics drop sharply during unlearning but are fully restored after relearning. This is observed in GA and RLabel with moderate hyperparameters. Besides, *reversible, non-catastrophic forgetting* entails only a mild, easily recoverable performance degradation, as seen with NPO at LR $= 3 \times 10^{-5}$, $N = 6$.

Conversely, *irreversible, catastrophic forgetting* occurs when relearning fails to restore utility, leaving F.Acc and R.Acc low despite partial MIA recovery. This pattern is common for GA and RLabel under aggressive hyperparameters (*e.g.*, LR $= 3 \times 10^{-5}$, $N = 100$), where cumulative updates lead to irreversible representational collapse. The MIA AUC metric behaves erratically in this regime: it may fall below 50% during unlearning but misleadingly rebounds to high values after relearning, even after the model's capabilities have been permanently lost. These empirical results on single and continual unlearning are consistent with the theoretical framework in Section 5, which shows that small perturbations to the model weights can trigger disproportionately large drops in accuracy.

Table 4: Single and continual unlearning results for GA across four models

| Single unlearning: Qwen2.5-7B (GA) | | | Continual unlearning: Qwen3-8B-Base (GA) | | | Continual unlearning: Llama-3-8B (GA) | | | Continual unlearning: Qwen2.5-3B (GA) | | |
|---|---|---|---|---|---|---|---|---|---|---|---|
| | MATH | GSM8K | | F.Acc | R.Acc | | F.Acc | R.Acc | | F.Acc | R.Acc |
| Original model | 9.00 | 80.10 | Original model | 78.28 | 62.96 | Original model | 76.41 | 63.50 | Original model | 76.37 | 61.39 |
| $3\times10^{-6}$ (unlearn) | 6.24 | 73.28 | $6\times10^{-6}$ (unlearn) | 0.45 | 0.21 | $6\times10^{-6}$ (unlearn) | 0.38 | 0.48 | $6\times10^{-6}$ (unlearn) | 1.45 | 2.56 |
| $3\times10^{-6}$ (relearn) | 8.97 | 78.29 | $6\times10^{-6}$ (relearn) | **79.72** | 62.66 | $6\times10^{-6}$ (relearn) | **76.49** | 63.21 | $6\times10^{-6}$ (relearn) | **79.61** | 61.45 |
| $6\times10^{-6}$ (unlearn) | 1.12 | 30.21 | $5\times10^{-5}$ (unlearn) | 0.02 | 0.02 | $5\times10^{-5}$ (unlearn) | 0.00 | 0.00 | $5\times10^{-5}$ (unlearn) | 0.01 | 0.01 |
| $6\times10^{-6}$ (relearn) | 8.62 | 77.63 | $5\times10^{-5}$ (relearn) | 0.03 | 0.03 | $5\times10^{-5}$ (relearn) | 0.02 | 0.04 | $5\times10^{-5}$ (relearn) | 3.58 | 4.27 |

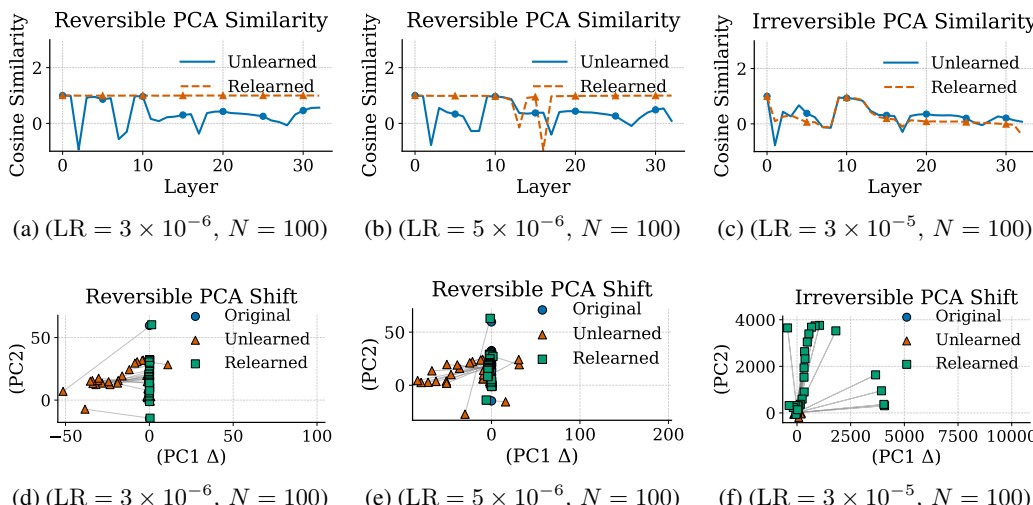

(a) $(\text{LR} = 3 \times 10^{-6}, N = 100)$  (b) $(\text{LR} = 5 \times 10^{-6}, N = 100)$  (c) $(\text{LR} = 3 \times 10^{-5}, N = 100)$

(d) $(\text{LR} = 3 \times 10^{-6}, N = 100)$  (e) $(\text{LR} = 5 \times 10^{-6}, N = 100)$  (f) $(\text{LR} = 3 \times 10^{-5}, N = 100)$

Figure 2: Layer-wise PCA Similarity and Shift for GA on Yi-6B (simple task). Vary LR $\{3 \times 10^{-6}, 5 \times 10^{-6}, 3 \times 10^{-5}\}$ at $N = 100$. Sustained low similarity or large shifts signal severe, irreversible catastrophic forgetting, whereas partial similarity or small shifts indicate mild, reversible catastrophic forgetting. Input queries are drawn from the forget set.

We conducted additional single-unlearning experiments on Qwen2.5-7B for the *complex task*, evaluating GA under two learning rates. We also performed continual-unlearning experiments ($N = 100$) on Llama-3-8B [6], Qwen2.5-3B [42], and Qwen3-8B-Base [43] for the *simple task*, again using GA with the same learning-rate settings. As shown in Table 4, the results further suggest that task-level metrics alone are insufficient to reliably assess the true reversibility of unlearning across different settings. The observed behaviors remain consistent with those in Table 3 and Table 2.

## 4 REPRESENTATION-LEVEL EVALUATION

### 4.1 REPRESENTATIONAL ANALYSIS TOOLS

To analyze representational drift, we employ four hidden state diagnostics, as summarized in Figure 1(b). Their precise definitions and implementation details are deferred to Appendix A.3.

**PCA Similarity, Shift, and Mean Distance.** For each layer $i$, we collect activation matrices $\mathbf{H}_i^{\text{orig}}$, $\mathbf{H}_i^{\text{unl}}$, and $\mathbf{H}_i^{\text{rel}}$ on a probe set $\mathcal{X}$ for the original, unlearned, and relearned models, respectively. Let $\mathbf{c}_{i,1}^{(*)}$ and $p_{i,12}^{(*)}$ be the first principal direction and its mean projection for state $(*) \in \{\text{orig}, \text{unl}, \text{rel}\}$. PCA Similarity is the cosine between $\mathbf{c}_{i,1}^{\text{orig}}$ and $\mathbf{c}_{i,1}^{(*)}$, while PCA Shift is the signed difference $p_{i,12}^{(*)}$. Small values for these metrics indicate stable features, whereas large, unrecovered shifts signify irreversible changes [49]. We also introduce the *mean PCA distance*, the average Euclidean distance on $p_{i,12}^{(*)}$ across layers, to provide a single scalar measure of representation drift.

**Centered Kernel Alignment (CKA).** Given centered activation matrices $X_i^{\text{orig}}$ and $Y_i^{(*)}$, we compute $\text{CKA}(X_i^{\text{orig}}, Y_i^{(*)}) \in [0, 1]$. Values $\approx 1$ mean nearly identical subspaces, those $\approx 0$ are orthogonal.
**Fisher information (FIM).** We estimate the diagonal of the empirical FIM by averaging squared

gradients over the probe set $\mathcal{X}$. Comparing $\text{FIM}^{\text{orig}}$, $\text{FIM}^{\text{unl}}$, and $\text{FIM}^{\text{rel}}$ reveals how unlearning alters the loss landscape and whether relearning restores parameter importance [16; 11].

All diagnostics are computed not only on the forget set but also on the retain set and unrelated data to distinguish targeted unlearning from general representational degradation.

## 4.2 REPRESENTATIONAL RESULTS

### 4.2.1 SINGLE UNLEARNING

Figure 3 demonstrates feature-level changes under single unlearning. (a) PCA Similarity remains near 1.0 across all layers, with minor, reversible dips, indicating that dominant activation directions are preserved. Slight dips in shallow and final layers are rapidly restored after relearning, suggesting minimal and reversible drift. (b) PCA shifts are minimal, and relearned representations closely realign with the original. (c) CKA values are nearly 1.0 for all model states, confirming that subspace structures remain intact. (d) FIM spectra show only mild, temporary shifts that are fully restored after relearning. These results, combined with the task-level evaluation in Section 3.2, demonstrate that single unlearning induces *reversible, non-catastrophic forgetting*. This highlights the limitation of classic (task-level) metrics, which fail to capture the superficial nature of the forgetting.

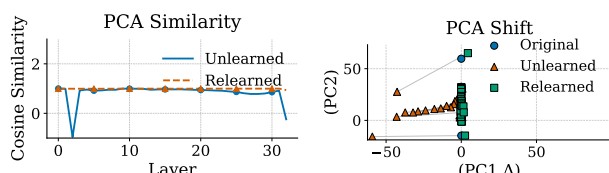

(a) Simple (Single Unlearning)  (b) Simple (Single Unlearning)

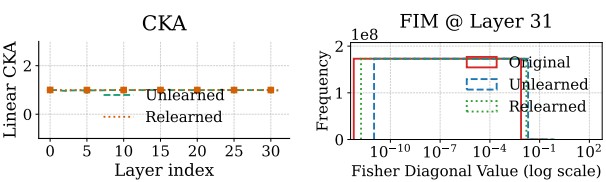

(c) Simple (Single Unlearning)  (d) Simple (Single Unlearning)

Figure 3: Single unlearning analysis on Yi-6B with GA under a simple task. In reversible non-catastrophic forgetting, PCA Similarity, PCA Shift, CKA, and FIM across layers show only minor changes with slight accuracy drops. Input queries are drawn from the forget set.

### 4.2.2 CONTINUAL UNLEARNING

As shown in Figures 2 and 5, PCA Similarity and Shift offer complementary views of representational change: similarity reflects global alignment, and shift is more sensitive to local variations. Relying on PCA similarity alone can obscure subtle effects; employing both avoids overlooking fine-grained distinctions, enabling a more comprehensive assessment. Higher learning rates or more requests cause sharp drops in similarity and large, unrecovered shifts, which is characteristic of *irreversible catastrophic forgetting*. In contrast, milder hyperparameters lead to high similarity and bounded shifts that are restored after relearning, consistent with *reversible, catastrophic forgetting*. This pattern is consistent across probe sets from forget set, retain set, and unrelated data (Figures 10 and 14).

Figure 4 integrates CKA (top) and FIM (bottom) analyses. CKA reveals that mild unlearning maintains stable alignment that recovers post-relearning, while aggressive unlearning causes irreversible degradation. The FIM spectra complement this by showing that continual unlearning flattens the loss landscape. Extreme hyperparameters induce a permanent leftward shift in sensitivity distributions, whereas moderate settings permit recovery. Together, these diagnostics suggest that observed performance loss is often due to temporary suppression rather than permanent erasure of knowledge. For conciseness, we present results on forget-set queries in the main text; retain-set queries, which yield similar conclusions under catastrophic forgetting, are in Appendix A.6 (*e.g.*, Figures 12 and 10).

**Mean PCA Distance Analysis.** To quantify representation-level drift with a single metric, we use the *mean PCA distance*. To assess the metric's sensitivity, we compute its mean and standard deviation across four random seeds, as well as shuffling the order of unlearning requests. As shown in Table 5, higher learning rates consistently increase the mean PCA distance for both unlearned and relearned models At a low learning rate (*e.g.*, $3 \times 10^{-6}$), the mean and variance of the distance remain low after relearning, indicating stable and reproducible recovery. In contrast, a high learning rate (*e.g.*, $3 \times 10^{-5}$) sharply increases both values, reflecting greater variability and incomplete recovery.

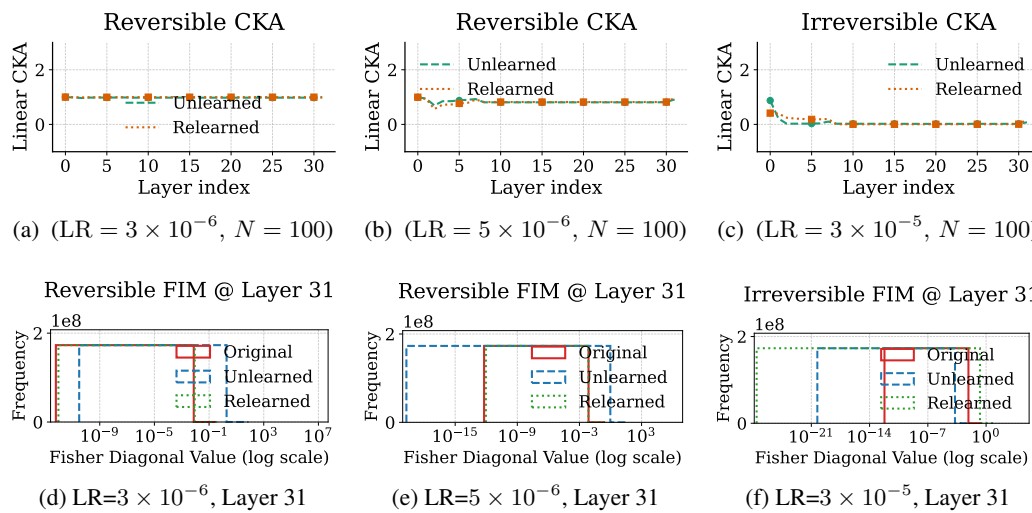

Figure 4: CKA for GA on Yi-6B, simple task. Vary LR $\{3 \times 10^{-6}, 5 \times 10^{-6}, 3 \times 10^{-5}\}$ with $N = 100$. High CKA ( 1) and concentrated FIM spectra indicates reversible catastrophic forgetting, while persistently low CKA and large-shifted, flattened spectra denote severe representational drift and irreversible catastrophic forgetting. Input queries are drawn from the forget set.

Table 5: Yi-6B (GA): Mean $\pm$ standard deviation of PCA distance on forget and retain sets across varying learning rates and random seeds; the number of unlearning requests is fixed.

| Model | Learning Rate | Phase | Seed | Mean PCA distance (forget set) | Mean PCA distance (retain set) |
|---|---|---|---|---|---|
| | $3 \times 10^{-6}$ | Unlearn | 12, 22, 32, 42 | $9.62 \pm 5.66$ | $6.85 \pm 4.10$ |
| | $3 \times 10^{-6}$ | Relearn | 12, 22, 32, 42 | $2.11 \pm 1.42$ | $1.64 \pm 1.12$ |
| | $5 \times 10^{-6}$ | Unlearn | 12, 22, 32, 42 | $11.52 \pm 6.19$ | $8.79 \pm 5.20$ |
| Yi-6B (GA) | $5 \times 10^{-6}$ | Relearn | 12, 22, 32, 42 | $1.37 \pm 0.74$ | $1.05 \pm 0.58$ |
| | $3 \times 10^{-5}$ | Unlearn | 12, 22, 32, 42 | $133.20 \pm 45.81$ | $121.45 \pm 38.60$ |
| | $3 \times 10^{-5}$ | Relearn | 12, 22, 32, 42 | $104.58 \pm 39.70$ | $95.34 \pm 32.40$ |

Combining these empirical findings with the task-level results in Section 3, we further connect to the theoretical analysis in Section 5, whose conclusions show that small weight perturbations induce only limited changes in the feature space, thereby enabling recovery, whereas larger perturbations accumulate into substantial representational drift that underlies irreversibility.

## 5 THEORETICAL ANALYSIS

### 5.1 A PERTURBATION MODEL OF UNLEARNING

To explain the empirical distinction between *reversible* and *irreversible* (catastrophic) forgetting, we introduce a perturbation model that links unlearning updates to representational changes across layers. Consider an $L$-layer feedforward network $f(x) = \sigma(W_L \, \sigma(\cdots \sigma(W_1 x) \cdots))$ with activation $\sigma$ and weights $W_{i\,i=1}^{L}$. We model unlearning as a layer-wise perturbation $\widetilde{W}_i = W_i + E_i$, where the magnitude of the error term, $|E_i| = O(\text{LR}, N)$, scales with the learning rate (LR) and the number of unlearning requests $N$. A Neumann-series expansion of the network's output shows that the total change $\widetilde{f}(x) - f(x)$ is defined as $\sum_{\emptyset \neq S \subseteq \{1,\ldots,L\}} (W_L \circ \cdots \circ E_{i_k} \circ \cdots \circ W_1)(x)$.

When perturbations are small and localized to a few layers, first-order terms dominate, leading to *reversible (catastrophic) forgetting*. In contrast, when comparable perturbations are distributed across many layers, higher-order interaction terms accumulate, causing structural degradation that results in *irreversible (catastrophic) forgetting*. We can formalize the impact on our diagnostic tools:

Table 6: Yi-6B (GA+GD+WAGLE) performance under different relearning settings. F.Acc/R.Acc are forget/retain accuracy, with mean PCA distances on the forget and retain sets.

| Phase | F.Acc | R.Acc | Mean PCA distance (forget set) | Mean PCA distance (retain set) |
|---|---|---|---|---|
| Original model | 78.9 | 65.5 | 0 | 0 |
| LR=$2 \times 10^{-5}$, $N = 50$, relearned by retain set ($N = 25$) | | | | |
| Unlearn | 37.8 | 55.9 | 11.84 | 6.28 |
| Relearn | 46.9 | 58.3 | 9.00 | 5.91 |
| LR=$4 \times 10^{-5}$, $N = 50$, relearned by unrelated data ($N = 50$) | | | | |
| Unlearn | 27.8 | 51.4 | 26.02 | 8.37 |
| Relearn | 31.5 | 53.5 | 24.56 | 8.12 |

Table 7: Relearning on Qwen3-8B-Base and Llama-3-8B (GA+GD+WAGLE) with unrelated data. F.Acc/R.Acc is forget/retain accuracy, with mean PCA distances on the forget and retain sets.

| | Qwen3-8B-Base (relearned by unrelated data) | | | | Llama-3-8B (relearned by unrelated data) | | | |
|---|---|---|---|---|---|---|---|---|
| Phase | F.Acc | R.Acc | PCA dist (forget) | PCA dist (retain) | F.Acc | R.Acc | PCA dist (forget) | PCA dist (retain) |
| Original model | 78.28 | 62.96 | 0.00 | 0.00 | 76.41 | 63.50 | 0.00 | 0.00 |
| Relearning by unrelated data (Qwen: LR=$5 \times 10^{-6}$, N=50; Llama: LR=$3 \times 10^{-6}$, N=50) | | | | | | | | |
| Unlearn | 48.52 | 56.47 | 8.49 | 5.98 | 42.59 | 53.47 | 14.29 | 7.12 |
| Relearn | 53.21 | 59.16 | 6.57 | 4.32 | 49.78 | 56.24 | 11.47 | 6.21 |

**PCA Similarity.** Let $X_i$ and $Y_i = X_i + E'_i$ be the centered activations at layer $i$ before and after unlearning. By the Davis–Kahan theorem [5], $\cos \angle(\mathbf{c}_i^{\text{orig}}, \mathbf{c}_i^{\text{upd}}) \approx 1 - O(\|E'_i\|/(\lambda_{1,i} - \lambda_{2,i}))$, with top two eigenvalues $\lambda_{1,i}, \lambda_{2,i}$. The layer-averaged PCA similarity is $\bar{S}_{\text{PCA}} \approx 1 - O((1/L) \sum_i \|E'_i\|)$.
**PCA Shift.** Along the first principal component, the activation-centroid shift is expressed as $p_{i,12} = O(\|E'_i\|)$. Large perturbations $\|E'_i\|$ propagating across multiple layers lead to *irreversible* representational drift, whereas smaller perturbations remain localized and thus *reversible*.
**CKA.** Let $\widetilde{K}_{Y_i} = \widetilde{K}_{X_i} + \Delta K_i$ denote the perturbed Gram matrix at layer $i$. The corresponding CKA score is computed as $\text{CKA}_i = 1 - O\left(\|\Delta K_i\|_*/\|\widetilde{K}_{X_i}\|_*\right)$. Averaging across layers yields $\bar{C} \approx 1 - O\left(\frac{1}{L} \sum_i \|\Delta K_i\|_*\right)$, where $\bar{C}$ denotes the layer-averaged CKA.
**Fisher Information.** Given update $\delta w_i = O(\|E_i\|)$, the Fisher diagonal behaves as $F_{ii}(w + \delta w) = F_{ii}(w) + O(\|\delta w_i\|)$, so the average Fisher becomes $\bar{F} = (1/P) \sum_i F_{ii} = F_0 - O((1/P) \sum_i \|E_i\|)$.

## 5.2 BRIDGING REPRESENTATIONAL DRIFT AND TASK-LEVEL METRICS

Classic (task-level) metrics can be misleading. They are highly sensitive to small weights' changes, particularly in the final layers, which can cause large shifts in output probabilities without altering the model's deeper representations. For a softmax output, a small perturbation $\delta\theta$ to the model parameters yields a large change in log-probability: $\log p(y|x; \theta + \delta\theta) \approx \log p(y|x; \theta) + \nabla_\theta \log p(y|x; \theta)^\top \delta\theta + O(\|\delta\theta\|^2)$. A minor update to the logits can dominate this first-order term, causing a sharp drop in accuracy that suggests catastrophic forgetting, even if the underlying geometry is preserved.

This aligns our theoretical model with the empirical findings in Sections 3 and 4.2. When LR or $N$ is small, changes are confined to first-order effects, feature spaces remain intact, and forgetting is *reversible*. When LR or $N$ is large, higher-order perturbations accumulate across layers, making recovery impossible and leading to *irreversible* forgetting. Figure 2 illustrates such a transition.

Interestingly, relearning can sometimes lead to performance that *exceeds* the original model's accuracy on the forget set (Table 3). This suggests that unlearning can act as a form of contrastive regularization, reinforcing salient features related to the forgotten data, which a brief relearning can then exploit.

## 5.3 PROBING THE LIMITS OF IRREVERSIBILITY

In our primary experiments, we did not observe *irreversible non-catastrophic forgetting*; even a small fraction (*e.g.*, 10%) of the forget set was sufficient to restore performance. To explore this regime, we conducted extra experiments with more constrained relearning conditions. We used the

GA+GD+WAGLE method [14], which selectively updates influential parameters, and limited the relearning data to either (i) $50\%$ of the retain set or (ii) an equal-sized, unrelated dataset (Table 6).

Under these conditions, the method exhibited *seemingly irreversible, non-catastrophic forgetting*. The forget set showed large, unrecoverable PCA distances, while the retain set experienced only modest, partially recoverable degradation. This demonstrates that achieving the ideal of targeted, permanent unlearning without collateral damage remains an open challenge. Defining precise thresholds to distinguish the forgetting regimes (*reversible vs. irreversible* and *catastrophic vs. non-catastrophic*) is non-trivial, as they depend on the unlearning method, task complexity, and other factors.

To assess generality, we apply GA+GD+WAGLE to Qwen3-8B-Base and Llama-3-8B, using an unrelated dataset for the relearning phase. Table 7 again shows *seemingly irreversible yet non-catastrophic forgetting*. We further find model-family differences in hyperparameter sensitivity when reaching comparable behavioral regimes. Such sensitivity likely shapes the boundary between reversible and irreversible forgetting and may guide future work on stable, irreversible, non-catastrophic unlearning.

## 6 DISCUSSION AND TAKEAWAYS

Beyond the diagnostic toolkit that identifies four regimes, representation-level signals provide a foundation for understanding, predicting, and guiding unlearning behaviour and also uncover the surprising possibility that unlearning can enhance performance rather than merely erase information.

**(1) Diagnostic metrics reliably predict reversibility under a fixed protocol.** Given a bounded relearning budget and fixed data source, large layer-wise PCA shifts and high mean PCA distance consistently predict recovery failure, whereas high CKA and concentrated Fisher spectra indicate reversibility. These thresholds remain stable across models, datasets, and unlearning methods.

**(2) Practical guidance for controlling unlearning behavior.** Practitioners can track mean PCA distance and CKA during unlearning to identify when to stop or adjust learning-rate and request budgets before entering irreversible collapse. Drift metrics help tune learning-rate schedules and request counts to deliberately target reversible or irreversible regimes. Fisher shifts and layer-wise drift localize which layers can be safely updated while preserving retain-set and unrelated representations.

**(3) Unlearning can enhance performance rather than merely erase information.** In several continual-unlearning runs, relearning on the forget set achieves accuracy that exceeds the original model. This indicates that unlearning behaves not only as a deletion mechanism but also as an implicit form of contrastive regularization. As discussed in Section 5, unlearning amplifies forget-specific subspaces, and relearning on augmented variants strengthens semantic structure and improves robustness, reorganizing representations toward more generalizable patterns.

## 7 CONCLUSION

This work demonstrates that class (task-level) evaluations of LLM unlearning are insufficient, as performance collapse often masks the reversibility of forgetting. Models may appear to have erased data while their internal representations remain intact and easily recoverable. Our representation-level toolkit reveals that genuine forgetting requires substantial, coordinated weight perturbations. Minor updates often create only a superficial, reversible effect. We find that achieving the ideal goal of irreversible, non-catastrophic forgetting remains an open challenge, exposing a fundamental limitation in current methods. Our findings call for a shift in evaluation, moving beyond surface-level metrics to protocols that measure true representational change. This is essential for developing unlearning algorithms that can provide meaningful and trustworthy guarantees of data removal.

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

# A APPENDIX

## A.1 LIMITATIONS

Our experiments target two LLMs and a handful of tasks and unlearning methods; although our diagnostic framework is model-agnostic and designed to scale, empirical validation on much larger models and production-scale pipelines remains to be done. The constrained relearning protocol and selected metrics provide clear insights into representational drift but are not exhaustive and do not offer formal privacy guarantees. In this work, we primarily relied on four diagnostic tools—PCA similarity, PCA shift, CKA, and Fisher information—to capture different aspects of representational drift. Other feature-level methods, such as correlation-based approaches (e.g., SVCCA [33]), offer similar perspectives on subspace similarity. Incorporating a broader suite of analytic tools is an important direction for future work.

## A.2 RELATED WORK

**Machine Unlearning.** Machine unlearning has emerged as a critical direction for addressing privacy, safety, and bias in large language models (LLMs) [44; 13; 32; 21; 24; 8; 36; 41; 48; 46; 40; 2]. It is typically defined as either *exact* or *approximate* [2]. Exact unlearning requires the resulting model to be indistinguishable from one retrained from scratch on the retain set, fully eliminating any statistical trace of the forget set. Approximate unlearning relaxes this requirement to distributional or behavioral similarity, demanding only comparable outputs (e.g., perplexity or forget-set accuracy) between unlearned and retrained models [29; 36]. For modern LLMs, however, exact unlearning is computationally infeasible, as full retraining or partition-based schemes scale poorly [2]. Consequently, approximate methods dominate practice in LLMs.

**Single-Shot Unlearning.** Most existing approaches are designed for single deletion events. Gradient-based strategies (e.g., GA) enforce forgetting directly but often incur significant utility loss [44]. Recent advances such as WAGLE augment these methods with weight attribution (e.g., GA+GD+WAGLE), selectively updating the most influential parameters to enhance forgetting efficacy while mitigating utility degradation [14]. Prompt-based steering avoids parameter updates, reducing cost, but typically achieves only superficial forgetting with vulnerability to reactivation [24]. Model-editing methods, such as AlphaEdit [22], are lightweight and potentially robust, yet their behavior under sequential or heterogeneous requests remains underexplored.

**Continual Unlearning.** When unlearning requests arrive sequentially, naive extensions of single-shot methods tend to compound damage, leading to catastrophic forgetting and unstable dynamics [1; 36]. Each request operates on an already modified model, magnifying utility loss. Recent work has attempted to mitigate this through orthogonal updates (e.g., LoRA-based unlearning [12]) and OOD detectors. ALKN [40] advances this line by providing a principled framework for continual unlearning, introducing parameter-level interventions and adaptive modules to counteract accumulative decline.

**Evaluations.** Evaluating unlearning efficacy remains an open challenge. Existing studies rely on three main classes of metrics. First, *classic (task-level) metrics* such as accuracy, perplexity [44; 21] are widely used but can be misleading, since performance degradation does not guarantee removal of knowledge. Second, *memorization probes* [18] assess verbatim recall, offering finer granularity but failing to capture semantic or paraphrased knowledge. Third, robustness-based evaluations examine vulnerabilities to *jailbreaking* [50; 25], *relearning attacks* [26], *prompt attack* [31] and even *quantization attacks* [48]. For *quantization attacks*, low-bit compression restores forget-set behavior without direct access to the forget. Lastly, RESTOR [34] evaluates whether an unlearning algorithm can both remove the influence of the forget set and restore the model to the parameter state it would have had if those datapoints had never been included in training. While existing approaches expose weaknesses in reversibility, they often conflate forgetting with class task-level degradation and lack structural insight. Our representation-level toolkit closes this gap by jointly diagnosing *reversibility* and *catastrophicity*, yielding a more faithful understanding of what is truly forgotten. We apply this toolkit to both single unlearning and continual unlearning, the latter of which has not been systematically investigated despite being a more realistic scenario where deletion requests arrive sequentially over a model's lifecycle.

## A.3 Detailed Analysis Tools

**PCA Similarity and PCA Shift.** For each Transformer layer, we perform PCA on the hidden activations of the *original* and *updated* models. Let $\mathbf{c}_{i,1}^{\text{orig}}$ and $\mathbf{c}_{i,1}^{\text{upd}}$ denote the first principal component (PC1) directions of layer $i$. The *PCA Similarity* is defined as

$$\text{PCA-Sim}(i) = \cos\left(\mathbf{c}_{i,1}^{\text{orig}}, \mathbf{c}_{i,1}^{\text{upd}}\right) = \frac{(\mathbf{c}_{i,1}^{\text{orig}})^\top \mathbf{c}_{i,1}^{\text{upd}}}{\|\mathbf{c}_{i,1}^{\text{orig}}\| \, \|\mathbf{c}_{i,1}^{\text{upd}}\|} \in [-1, 1],$$

where values near 1 indicate stable directional alignment, and values near $-1$ suggest a near-orthogonal shift in dominant directions.

To capture translational drift, we also compute the mean projection of activations along PC1 and PC2:

$$PC1\,\Delta(i) = \mathbf{c}_{i,1}^{\text{upd}} - \mathbf{c}_{i,1}^{\text{orig}}, \qquad PC2(i) = \mathbf{c}_{i,2}^{\text{upd}}, \qquad p_{i,12} = (PC1\,\Delta(i), PC2(i))$$

where $p_{i,12}$ quantifies displacement along PC1 and captures orthogonal deviation along PC2. These metrics reflect how the representation center drifts within the top subspace.

**Centered Kernel Alignment (CKA).** To assess subspace alignment, we use linear Centered Kernel Alignment (CKA) [17], which compares activation matrices $X, Y \in \mathbb{R}^{N \times D}$ from before and after unlearning. First, we compute the centered Gram matrices:

$$\widetilde{K}_X = H X X^\top H, \qquad \widetilde{K}_Y = H Y Y^\top H, \qquad H = I_N - \frac{1}{N}\mathbf{1}\mathbf{1}^\top.$$

The CKA score is then given by:

$$\text{CKA}(X, Y) = \frac{\text{Tr}(\widetilde{K}_X \widetilde{K}_Y)}{\sqrt{\text{Tr}(\widetilde{K}_X^2)}\sqrt{\text{Tr}(\widetilde{K}_Y^2)}} \in [0, 1],$$

where values near 1 indicate highly overlapping subspaces, and values near 0 signal near-orthogonality.

**Fisher Information.** To measure parameter-level importance, we compute the diagonal of the empirical Fisher Information Matrix (FIM). For each parameter $w_i$ and input distribution $\mathcal{D}_{\text{dis}}$, the diagonal entry is approximated as:

$$\text{FIM}_{ii} \approx \mathbb{E}_{(\mathbf{x},y)\sim\mathcal{D}_{\text{dis}}}\left[\left(\partial_{w_i} \log p(y \mid \mathbf{x}; \mathbf{w})\right)^2\right].$$

Larger values indicate that $w_i$ has a stronger influence on the model's predictions. A substantial leftward shift in the Fisher spectrum after unlearning implies a flattened loss landscape and diminished parameter sensitivity.

Together, these tools form a feature-space diagnostic suite: FIM captures global sensitivity, CKA measures subspace preservation, and PCA-based metrics expose fine-grained geometric drift across layers—enabling a robust assessment of representational degradation during unlearning.

## A.4 Different types of relearning and Sample efficiency

### A.4.1 Different types of relearning

Beyond standard relearning, we further evaluated the unlearned Yi-6B model (GA-based setup) under four alternative recovery strategies: quantization attacks [48], prompt attacks [31], jailbreaking [25], and in-context recovery. For quantization, we applied Int4 quantization directly to the unlearned model. For the other methods, which do not modify parameters, we adapted inputs to interface with our PCA analysis: *prompt attack* used paraphrased variants of the original inputs; *jailbreak attack* prepended the fixed prefix from [25]; *in-context recovery* supplied five demonstrations from the forget set before evaluating the original inputs.

As shown in Table 11, none of these recovery strategies restore the forgotten knowledge. Once the model enters the regime of reversible catastrophic forgetting, methods that do not explicitly update parameters or introduce only minor perturbations (quantization) fail to recover lost representations. This demonstrates that explicit relearning is necessary to reverse this particular forgetting state.

Table 8: **Yi-6B simple-task metrics under four** $(\mathrm{LR}, N)$ **settings.** For each block: forget/retain perplexity (F.Ppl / R.Ppl), forget/retain accuracy (F.Acc / R.Acc), CommonsenseQA (CSQA), GSM8K, and membership-inference AUC (MIA).

| Phase | Method | F.Ppl | R.Ppl | F.Acc | R.Acc | CSQA | GSM8K | MIA |
|---|---|---|---|---|---|---|---|---|
| \multicolumn{9}{c}{LR=$3 \times 10^{-5}$, $N=100$} |
| Original | — | 3.8 | 7.8 | 78.9 | 65.5 | 73.1 | 39.6 | 70.9 |
| | GA | $\infty$ | $\infty$ | 0.0 | 0.0 | 19.3 | 0.0 | 26.1 |
| | GA+GD | $\infty$ | $\infty$ | 9.7 | 2.3 | 19.7 | 0.0 | 16.8 |
| Unlearn | GA+KL | $\infty$ | $\infty$ | 9.0 | 6.2 | 19.6 | 0.0 | 17.8 |
| | NPO | 31296.5 | 597.9 | 37.8 | 37.9 | 62.2 | 1.0 | 60.1 |
| | NPO+KL | 348080.2 | 4482.0 | 64.3 | 55.9 | 64.9 | 1.4 | 59.0 |
| | Rlable | 63791.7 | 65903.4 | 0.0 | 0.0 | 20.9 | 0.0 | 65.1 |
| | GA | 137094.5 | 758443.5 | 2.1 | 1.8 | 19.7 | 0.0 | 74.5 |
| | GA+GD | 5274.5 | 9568.6 | 2.2 | 2.6 | 19.6 | 0.0 | 68.1 |
| Relearn | GA+KL | 5037.1 | 15019.9 | 1.7 | 1.6 | 20.6 | 0.0 | 70.7 |
| | NPO | 16.6 | 41.7 | 57.0 | 45.6 | 51.8 | 0.6 | 70.0 |
| | NPO+KL | 21.8 | 16.2 | 60.7 | 54.3 | 48.0 | 0.9 | 67.7 |
| | Rlable | 4056.1 | 15048.6 | 4.3 | 2.8 | 19.7 | 0.0 | 69.5 |
| \multicolumn{9}{c}{LR=$5 \times 10^{-6}$, $N=100$} |
| | GA | $\infty$ | $\infty$ | 9.1 | 6.2 | 19.6 | 0.0 | 23.2 |
| | GA+GD | $\infty$ | $\infty$ | 3.6 | 3.1 | 24.5 | 0.0 | 28.7 |
| Unlearn | GA+KL | $\infty$ | $\infty$ | 9.1 | 6.2 | 19.6 | 0.0 | 27.3 |
| | NPO | 3017.7 | 1110.6 | 50.1 | 52.3 | 72.9 | 37.5 | 50.6 |
| | NPO+KL | 38.5 | 232.4 | 77.6 | 64.3 | 73.1 | 37.6 | 65.4 |
| | Rlable | 57035.4 | 53377.1 | 0.1 | 0.4 | 19.1 | 0.0 | 63.6 |
| | GA | 3.7 | 7.8 | 80.0 | 64.9 | 70.2 | 39.9 | 68.0 |
| | GA+GD | 3.6 | 7.6 | 81.2 | 65.1 | 72.1 | 39.0 | 69.8 |
| Relearn | GA+KL | 3.6 | 8.4 | 81.1 | 64.8 | 71.6 | 40.7 | 68.3 |
| | NPO | 3.5 | 7.6 | 82.7 | 65.5 | 74.0 | 39.7 | 68.0 |
| | NPO+KL | 3.5 | 7.8 | 83.8 | 65.6 | 74.1 | 39.7 | 69.5 |
| | Rlable | 3.6 | 7.7 | 80.8 | 65.3 | 71.8 | 39.2 | 70.3 |
| \multicolumn{9}{c}{LR=$3 \times 10^{-6}$, $N=100$} |
| | GA | $\infty$ | $\infty$ | 16.8 | 14.4 | 69.5 | 12.3 | 25.2 |
| | GA+GD | 3.3 | 7.6 | 78.8 | 65.5 | 77.0 | 37.5 | 69.4 |
| Unlearn | GA+KL | $\infty$ | $\infty$ | 35.4 | 40.6 | 63.2 | 18.3 | 18.9 |
| | NPO | 3.7 | 7.9 | 78.3 | 65.0 | 73.3 | 38.7 | 68.4 |
| | NPO+KL | 3.8 | 8.1 | 78.4 | 65.1 | 73.6 | 38.6 | 66.7 |
| | Rlable | 36794.7 | 32562.0 | 3.8 | 3.2 | 19.3 | 2.2 | 61.4 |
| | GA | 3.7 | 7.6 | 80.8 | 65.2 | 73.4 | 39.9 | 68.6 |
| | GA+GD | 3.6 | 7.4 | 81.8 | 65.5 | 72.1 | 39.0 | 70.0 |
| Relearn | GA+KL | 3.6 | 10.3 | 81.0 | 63.3 | 67.2 | 40.7 | 70.7 |
| | NPO | 3.5 | 7.5 | 81.2 | 65.4 | 72.9 | 39.7 | 69.9 |
| | NPO+KL | 3.5 | 7.5 | 83.8 | 65.5 | 73.0 | 39.7 | 69.9 |
| | Rlable | 3.6 | 7.6 | 80.5 | 65.3 | 72.2 | 39.2 | 70.0 |
| \multicolumn{9}{c}{LR=$3 \times 10^{-5}$, $N=6$} |
| | GA | inf | inf | 36.3 | 36.1 | 69.1 | 5.8 | 29.6 |
| | GA+GD | 209.3 | 20.6 | 77.0 | 64.0 | 70.0 | 37.8 | 66.9 |
| Unlearn | GA+KL | inf | inf | 53.0 | 41.5 | 68.3 | 2.0 | 29.5 |
| | NPO | 12.3 | 10.7 | 71.6 | 59.4 | 71.7 | 24.7 | 68.7 |
| | NPO+KL | 8.9 | 10.7 | 74.7 | 62.1 | 72.8 | 32.2 | 67.9 |
| | Rlable | 51589.2 | 40622.9 | 0.4 | 0.7 | 19.8 | 0.0 | 62.6 |
| | GA | 6.8 | 11.4 | 70.5 | 58.7 | 64.5 | 18.4 | 68.2 |
| | GA+GD | 12.3 | 11.5 | 61.6 | 54.4 | 61.3 | 7.3 | 67.1 |
| Relearn | GA+KL | 17.1 | 11.6 | 66.6 | 56.2 | 60.6 | 3.0 | 65.0 |
| | NPO | 6.0 | 11.6 | 71.2 | 59.4 | 59.4 | 2.0 | 68.4 |
| | NPO+KL | 7.3 | 11.6 | 67.6 | 56.1 | 42.9 | 1.6 | 69.0 |
| | Rlable | 6.4 | 11.4 | 72.7 | 61.1 | 67.5 | 28.9 | 65.2 |

### A.4.2 SAMPLE EFFICIENCY

To examine sample efficiency, we extend our GA-based relearning experiments ($LR = 6 \times 10^{-6}$, $N = 100$) across three data sources—the forget set, retain set, and unrelated data (see Section 2 for details). Each source is evaluated at 10%, 30%, 60%, and 100% of the original forget-set size.

As shown in Table 12, these experiments reveal a clear hierarchy in recovery efficiency. Relearning on the forget set provides the strongest and fastest recovery, with PCA distances approaching those of the

Table 9: Qwen-2.5-7B: MIA / MATH / GSM8K Accuracy (%) for complex task under four settings. Bold numbers indicate improvements over the Original baseline in MATH or GSM8K.

| Phase | Method | LR=$3 \times 10^{-5}$, $N$=6 | | | LR=$3 \times 10^{-6}$, $N$=6 | | | LR=$5 \times 10^{-6}$, $N$=6 | | | LR=$5 \times 10^{-6}$, $N$=100 | | |
|---|---|---|---|---|---|---|---|---|---|---|---|---|---|
| | | MIA | MATH | GSM8K | MIA | MATH | GSM8K | MIA | MATH | GSM8K | MIA | MATH | GSM8K |
| Original | —— | 99.3 | 9.0 | 80.1 | 99.3 | 9.0 | 80.1 | 99.3 | 9.0 | 80.1 | 99.3 | 9.0 | 80.1 |
| Unlearn | GA | 5.9 | 0.0 | 0.0 | 0.9 | 0.0 | 0.0 | 3.8 | 0.0 | 0.0 | 5.5 | 0.0 | 0.0 |
| | NPO | 95.9 | 0.0 | 0.2 | 97.4 | 21.5 | 74.1 | 67.4 | 24.1 | 71.8 | 94.7 | 0.0 | 0.4 |
| | RLabel | 35.5 | 0.0 | 0.0 | 69.6 | 0.0 | 1.5 | 11.2 | 0.0 | 0.0 | 2.9 | 0.0 | 0.0 |
| Relearn | GA | 97.6 | 0.0 | 1.1 | 99.3 | 5.1 | **83.2** | 99.4 | **9.3** | 77.8 | 99.2 | 0.0 | 0.0 |
| | NPO | 95.8 | 0.0 | 0.0 | 99.4 | 4.7 | **82.6** | 99.4 | **16.5** | 75.7 | 99.2 | 0.0 | 0.0 |
| | RLabel | 99.5 | 0.0 | 0.0 | 99.3 | 5.3 | **83.3** | 99.3 | **10.0** | 77.2 | 99.6 | 0.0 | 0.0 |

Table 10: Yi-6B (GA): Mean PCA distance under different learning rates. The left block uses China Taiwan for *relearning* only, while the right block uses TOFU for both *unlearning* and *relearning*.

| | Relearning with China Taiwan | | Unlearning + Relearning with TOFU | |
|---|---|---|---|---|
| Learning Rate | Phase | Mean PCA distance (forget set) | Phase | Mean PCA distance (forget set) |
| $3 \times 10^{-6}$ | Unlearn | 17.12 | Unlearn | 0.51 |
| $3 \times 10^{-6}$ | Relearn | 4.98 | Relearn | 0.27 |
| $5 \times 10^{-6}$ | Unlearn | 20.27 | Unlearn | 2.41 |
| $5 \times 10^{-6}$ | Relearn | 10.77 | Relearn | 1.08 |
| $3 \times 10^{-5}$ | Unlearn | 193.13 | Unlearn | 11.96 |
| $3 \times 10^{-5}$ | Relearn | 167.32 | Relearn | 11.02 |

original model even at moderate sample sizes. In contrast, relearning using the retain set or unrelated data restores performance only gradually; both sources are substantially less sample-efficient and yield slower improvements in representational alignment.

## A.5 MEAN PCA DISTANCE UNDER DIFFERENT DATASET

To examine the role of distributional alignment, we evaluate unlearning and relearning under two dataset settings. First, we use the *TOFU* benchmark [29], where both unlearning and relearning occur within the same distribution. Second, treating different languages as out-of-distribution (OOD), we include a Traditional-Chinese corpus for relearning. This setup enables us to probe whether cross-lingual signals can drive effective recovery, and how their efficacy compares with in-distribution.

Table 10 confirms that **cross-lingual relearning improves the model but achieves less complete restoration than English data**: mean PCA distance and related summary metrics move closer to baseline values, yet remain substantially higher. Greater linguistic or domain dissimilarity therefore reduces the efficacy of recovery, though partial restoration is still attainable.

For the TOFU dataset, the overall pattern holds: **learning rate and the number of unlearning requests ($N$) effectively regulate feature drift and reversibility**. However, the representational shifts induced by TOFU are milder than those observed in our simple and complex tasks. We attribute this to the smaller and less diverse nature of TOFU's corpus; many entries are short and contain only author metadata, making its impact on the model's feature space comparatively limited.

## A.6 DETAILED ANALYSIS RESULTS

### A.6.1 PRINCIPAL COMPONENT ANALYSIS: SIMILARITY AND SHIFT

Across the same hyper-parameter grid, Figure 7 (PCA–Similarity) and Figure 11 (PCA–Shift) provide complementary views of representational drift. For GA, higher learning rates drive unlearned states (orange) far from the original (blue), while relearning (green) fails to return, producing long rays of *irreversible* drift. GA+GD narrows the spread but still collapses at $3 \times 10^{-5}$.

On Qwen-2.5-7B, GA shifts span thousands of PC1 units and drive PC2 to extreme negatives (Figure 13c,f,i), consistent with the multi-layer perturbations predicted in Section 4. In complex tasks

Table 11: Different recovery attempts on Yi-6B (GA, LR=$6 \times 10^{-6}$, $N = 100$). F.Acc and mean PCA distance are computed on the forget set.

| Setting (Yi-6B, GA, LR=$6 \times 10^{-6}$, $N = 100$) | F.Acc | Mean PCA distance (forget set) |
|---|---|---|
| Original model | 78.90 | 0.00 |
| Unlearned model | 0.00 | 31.66 |
| Quantization attack | 0.00 | 32.21 |
| In-context (num_demos = 5) | 0.01 | 30.83 |
| Prompt attack | 0.03 | 29.14 |
| Jailbreaking | 0.03 | 30.04 |

Table 12: Relearning comparison on Yi-6B (GA, LR=$6 \times 10^{-6}$, $N = 100$), evaluating the sample efficiency of different relearning data sources (forget, retain, unrelated). The results show how varying the amount and type of relearning data affects recovery performance and representational drift.

| Setting (Yi-6B, GA, LR=$6 \times 10^{-6}$, $N = 100$) | F.Acc | Mean PCA distance (forget set) |
|---|---|---|
| Original model | 78.90 | 0.00 |
| Unlearned model | 0.00 | 31.66 |
| **Relearned by forget set** | | |
| 10% | 67.28 | 8.49 |
| 30% | 75.77 | 6.42 |
| 60% | 77.13 | 4.31 |
| 100% | 79.20 | 2.16 |
| **Relearned by retain set** | | |
| 10% | 0.05 | 30.57 |
| 30% | 11.24 | 25.48 |
| 60% | 45.24 | 14.69 |
| 100% | 75.86 | 7.51 |
| **Relearned by unrelated data** | | |
| 10% | 0.02 | 31.02 |
| 30% | 6.48 | 27.74 |
| 60% | 38.83 | 17.51 |
| 100% | 65.66 | 9.14 |

such as mathematical reasoning, even small perturbations in hidden states can lead to substantial performance differences. This is reflected in our PCA–Similarity analysis, where seemingly minor changes in hidden state geometry correspond to meaningful behavioral variations. Besides, PCA-Similarity captures global alignment, whereas PCA–Shift highlights fine-grained translational drift. This distinction also explains why Figure 9h,i show only moderate misalignment under similarity but reveal pronounced displacements under shift (cf. Figure 13). Using both metrics thus provides a more complete characterization of reversibility. Overall, these results confirm that GA, with or without GD or KL, induces large and often irreversible displacements, whereas NPO variants, and to a lesser extent RLabel, constrain less shifts, consistent with our utility findings.

### A.6.2 CENTERED KERNEL ALIGNMENT ANALYSIS

Figures 15–17 report layer-wise linear CKA between the original model and its unlearned or relearned counterparts. Across both Yi-6B and Qwen-2.5-7B, GA stands out: as the learning rate or $N$ increases, its CKA curve drops close to zero in most layers and fails to recover, revealing a deep subspace fracture consistent with the irreversible PCA trends. GA+GD and GA+KL mitigate this decline to some extent but do not restore full alignment after relearning.

Task complexity does not alter the ordering but amplifies the differences. On the math-heavy Qwen benchmark, GA's tail layers fall almost to zero at high learning rates, whereas NPO maintains significantly higher alignment. Taken together with the PCA-Shift results, these findings show that GA-style objectives consistently break subspace alignment, NPO variants preserve much greater stability, and RLabel induces moderate but partly recoverable distortions.

### A.6.3 FISHER INFORMATION ANALYSIS

Figures 19–33 plot the empirical Fisher spectra layer by layer. Across both Yi-6B (simple) and Qwen-2.5-7B (complex), GA and its variants exhibit a pronounced leftward shift of the diagonal

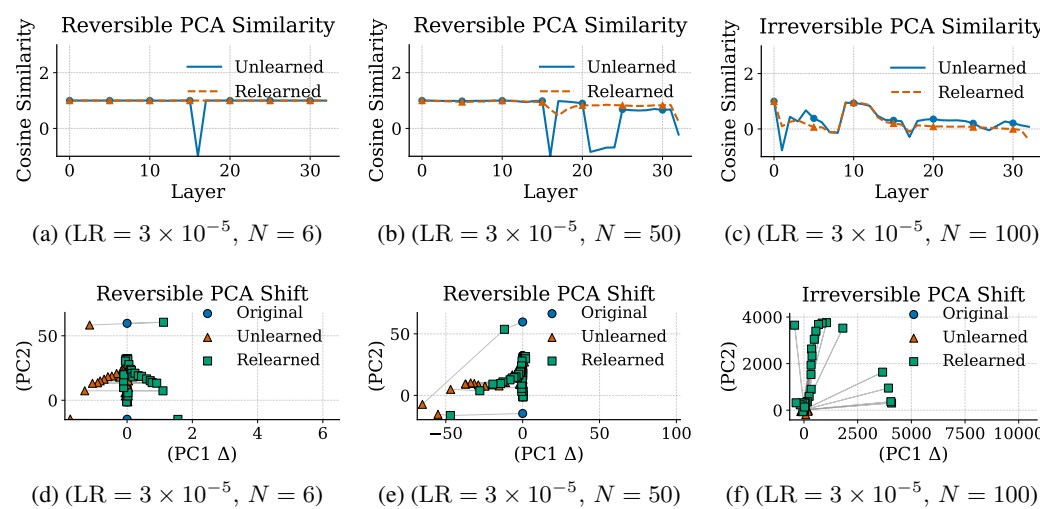

Figure 5: Layer-wise PCA Similarity and Shift for GA on Yi-6B (simple task). vary $N \in \{6, 50, 100\}$ at $LR = 3 \times 10^{-5}$. Sustained low similarity or large shifts signal severe, irreversible catastrophic forgetting, whereas partial similarity or small shifts indicate mild, reversible catastrophic forgetting. Input queries are drawn from the forget set.

histogram as LR or $N$ increase. The peaks move several orders of magnitude in middle and deep layers, reflecting a flattened loss surface and diminished parameter salience. Crucially, these shifts persist after relearning, marking the onset of irreversible forgetting.

NPO, NPO+KL, and RL produce smaller leftward displacements under moderate LR or $N$, and their Fisher spectra recenter after relearning, indicating primarily reversible drift. Under extreme settings (e.g., $LR = 3 \times 10^{-5}$ or $N = 100$), these methods also show persistent displacement in some layers, suggesting milder but still irreversible forgetting.

Figures 14, 10, 18, and 34 examine relearning dynamics when the fine-tuning data and input query are drawn from the forget set, the retain set, or an unrelated data: i) across all sources, the overall trends are similar: alignment can be partially restored, but recovery is consistently weaker with unrelated data, underscoring that effective relearning depends on both the size and the relevance of the training set; ii) the observed behavior also varies with the choice of input queries. In the case of *reversible catastrophic forgetting*, all forget set, retain set, and unrelated data undergo the similar feature drifts.

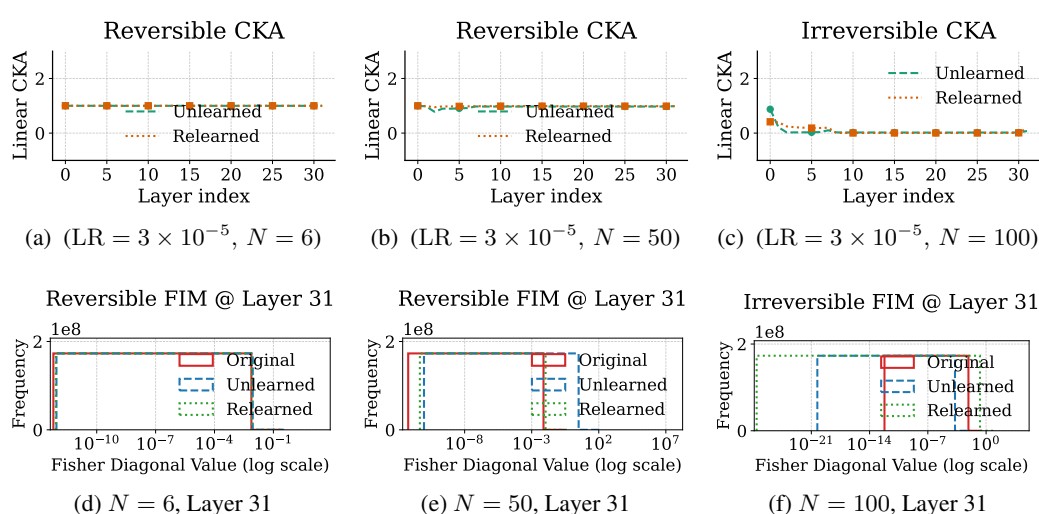

Figure 6: CKA and FIM for GA on Yi-6B, simple task. Vary LR $= 3 \times 10^{-5}$ with $N \in \{6, 50, 100\}$. High CKA ( 1) and concentrated FIM spectra indicates reversible catastrophic forgetting, while persistently low CKA and large-shifted, flattened spectra denote severe representational drift and irreversible catastrophic forgetting. Input queries are drawn from the forget set.

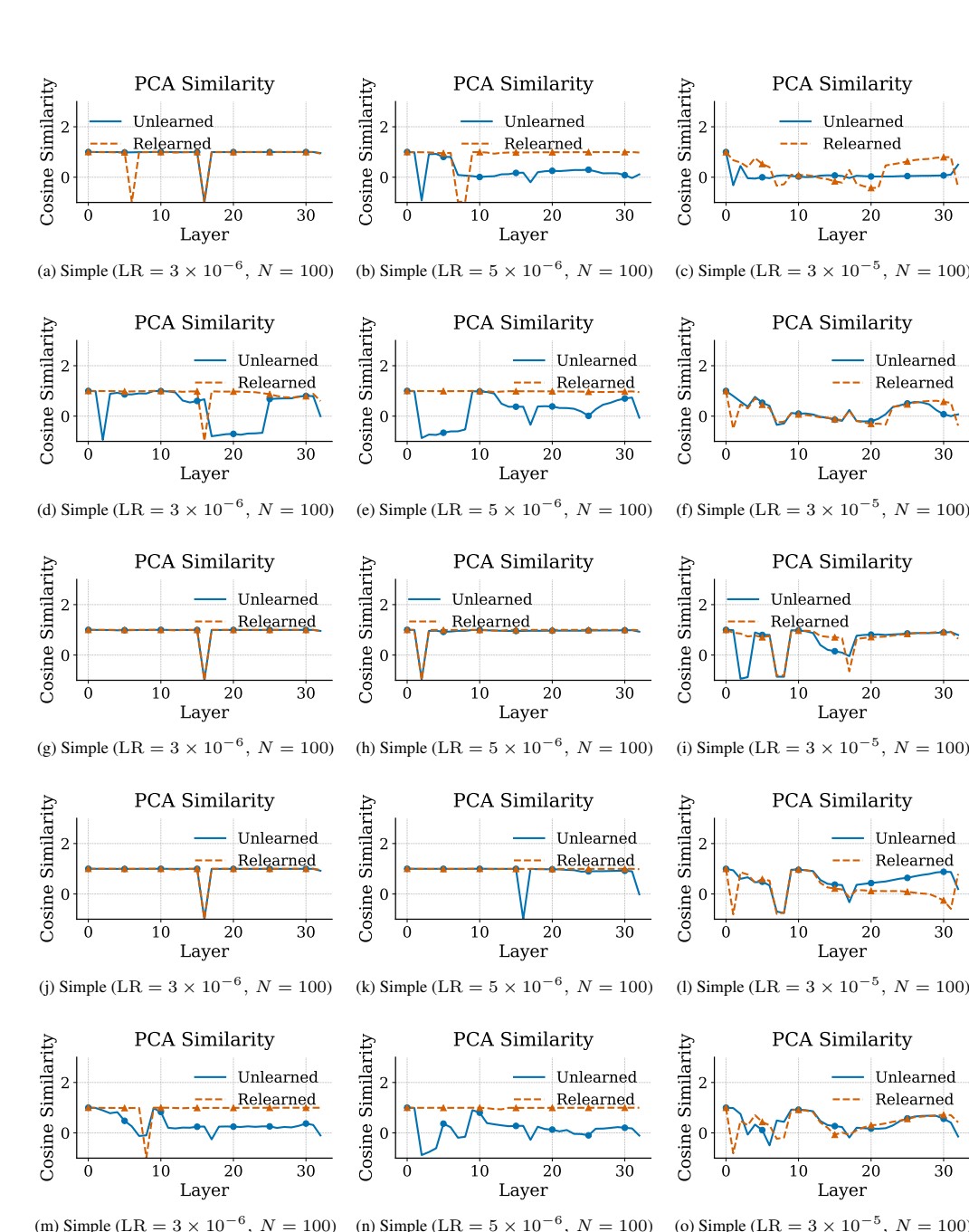

Figure 7: PCA Similarity Across Layers. Each row shows results under different unlearning methods: GA+GD (a–c), GA+KL (d–f), NPO (g–i), NPO+KL (j–l), and Rlable (m–o). All plots are for the simple task on Yi-6B, using three learning rates $\{3 \times 10^{-6}, 5 \times 10^{-6}, 3 \times 10^{-5}\}$ and fixed $N = 100$.

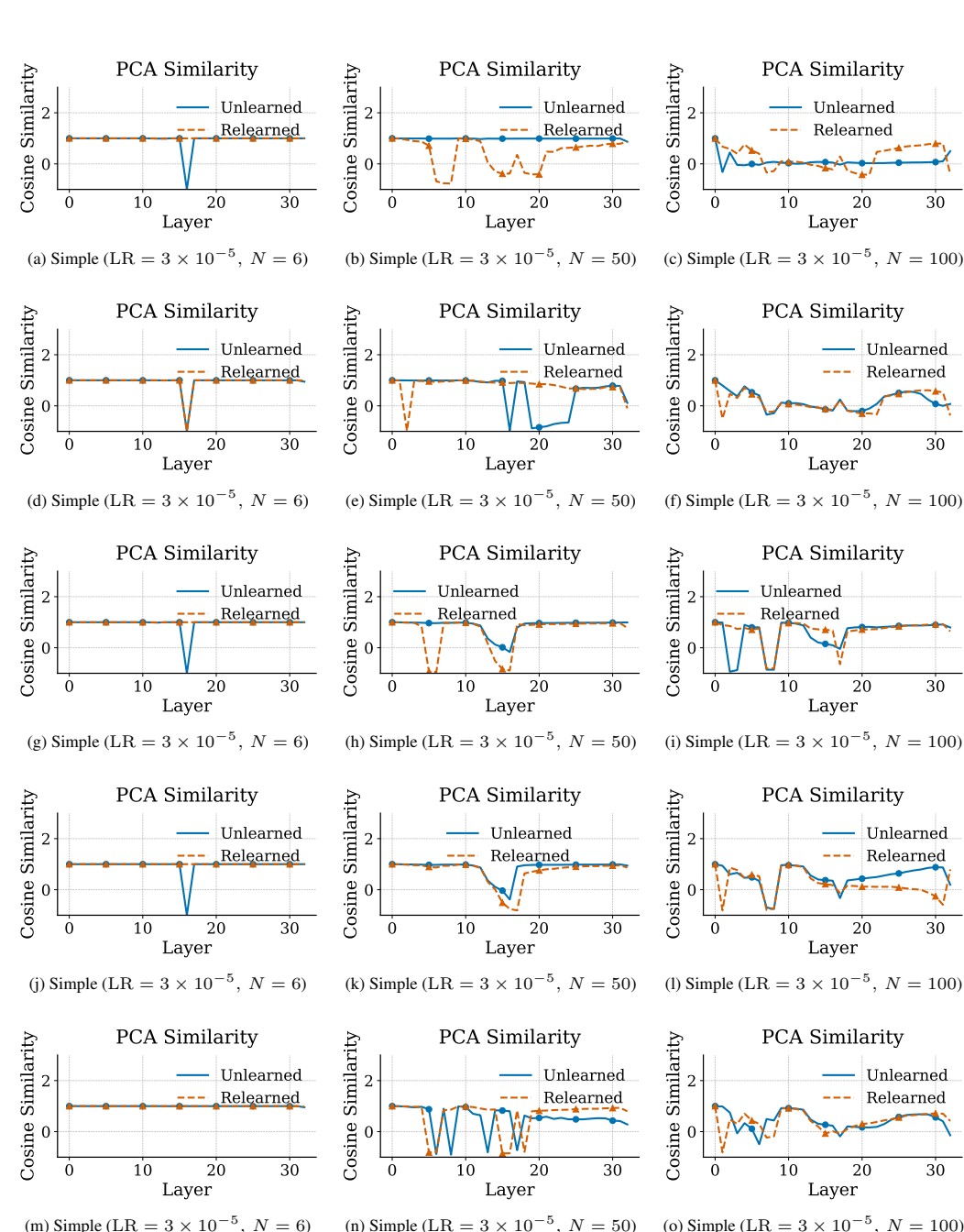

Figure 8: PCA Similarity Across Layers. Each row shows results under different unlearning methods: GA+GD (a–c), GA+KL (d–f), NPO (g–i), NPO+KL (j–l), and Rlable (m–o). Simple task on Yi-6B with fixed learning rate LR = $3 \times 10^{-5}$ and varying unlearning requests $N \in \{6, 50, 100\}$.

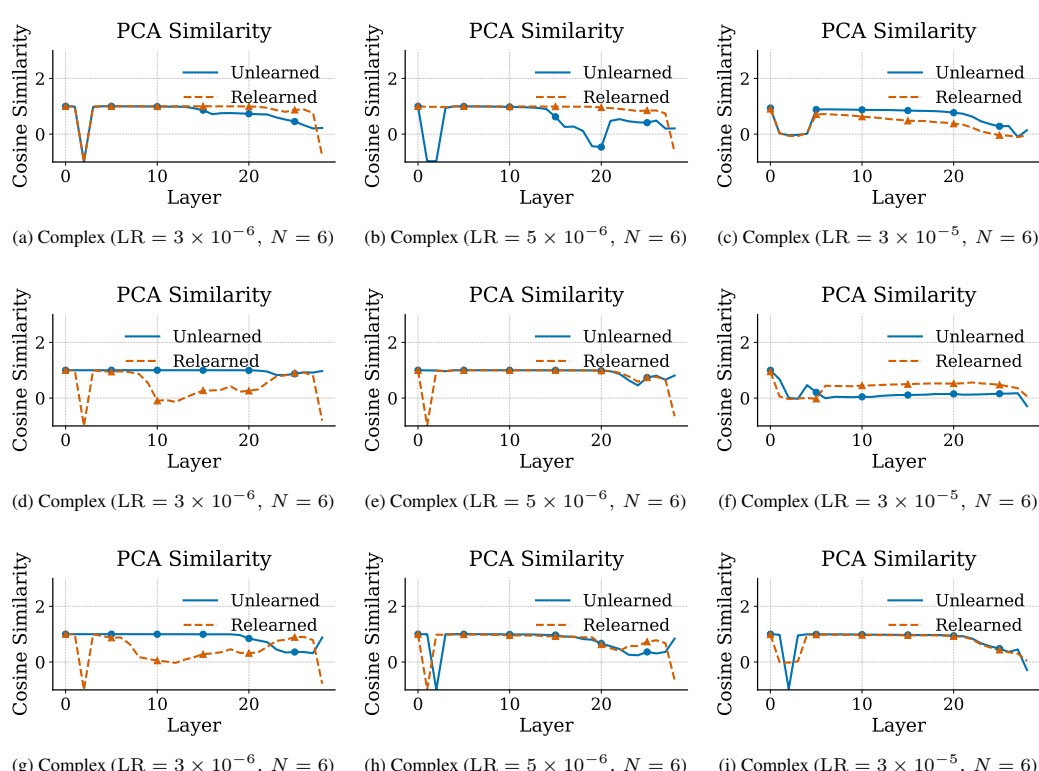

Figure 9: PCA Similarity Across Layers. Each row shows results under different unlearning methods: GA (a-c) NPO (d–f), Rlable (g–j). All plots are for the complex task on Qwen2.5-7B, using three learning rates $\{3 \times 10^{-6}, 5 \times 10^{-6}, 3 \times 10^{-5}\}$ and fixed $N = 6$.

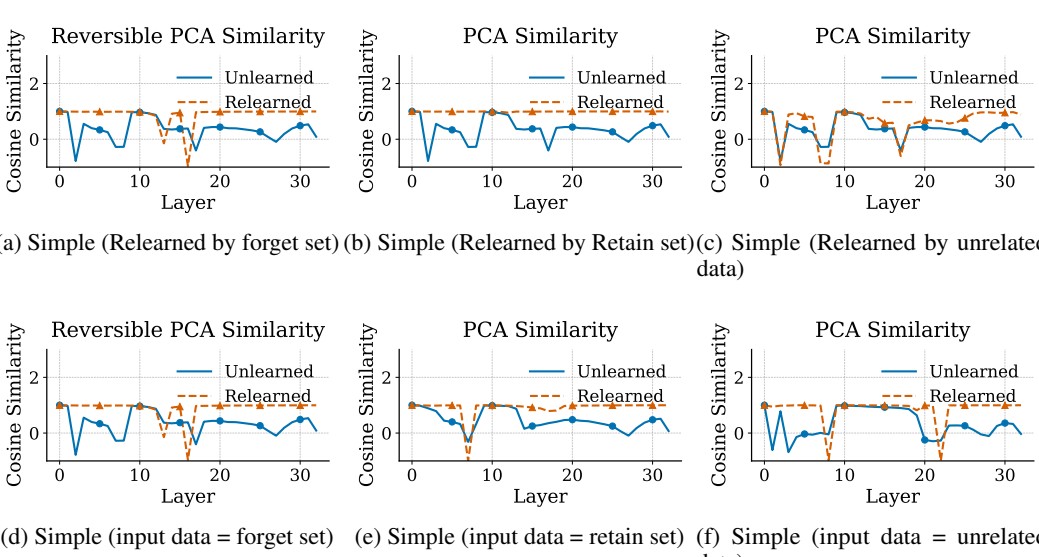

Figure 10: PCA Similarity Analysis for GA under Varied Relearning and Evaluation Inputs on Yi-6B (Simple Task). (a–c): Relearning is performed using the forget set, retain set, or unrelated data respectively. (d–f): PCA similarity is measured using the forget set, retain set, or unrelated data as evaluation input.

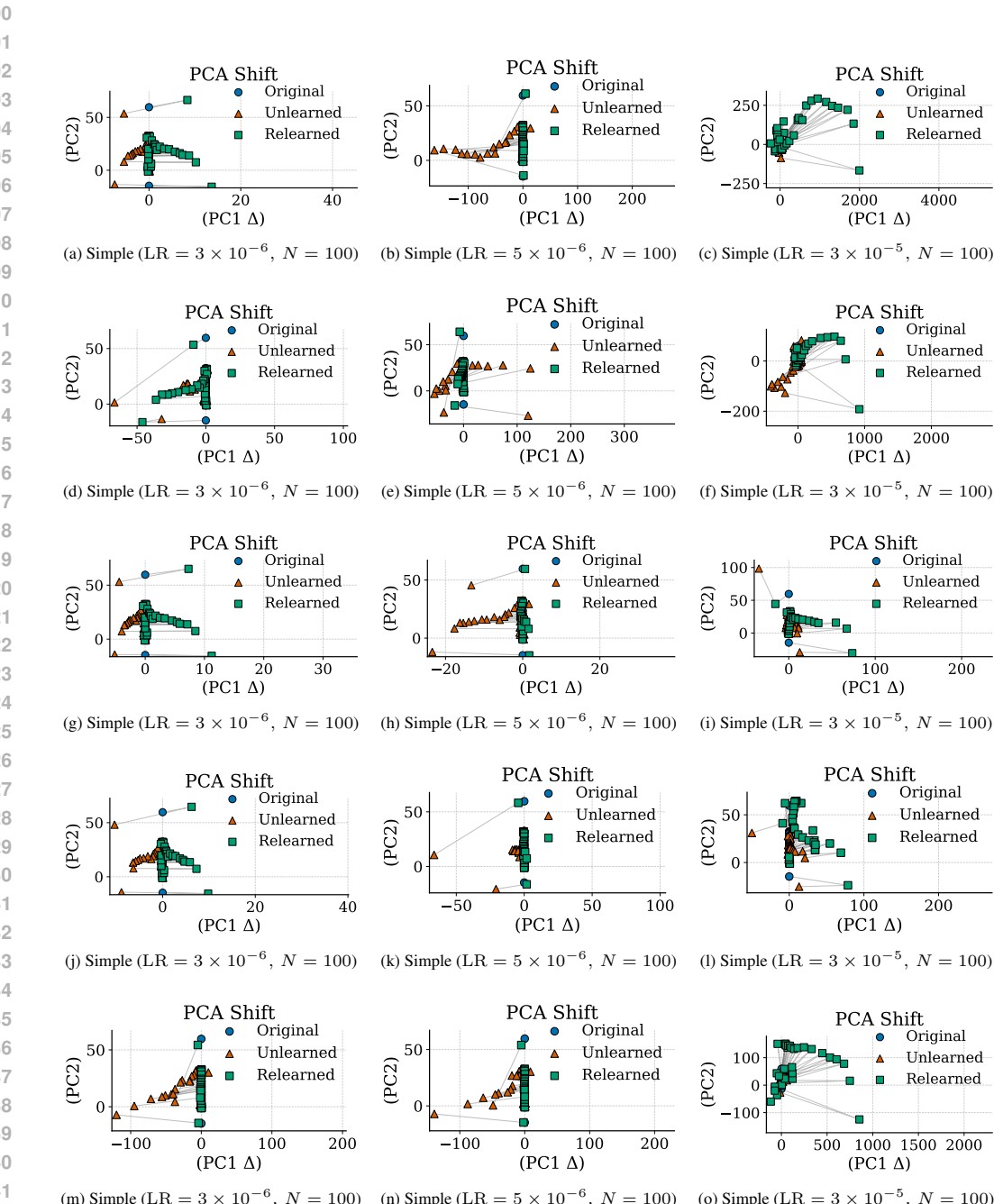

Figure 11: PCA Shift Across Layers. Each row shows results under different unlearning methods: GA+GD (a–c), GA+KL (d–f), NPO (g–i), NPO+KL (j–l), and Rlable (m–o). All plots are for the simple task on Yi-6B, using three learning rates $\{3 \times 10^{-6}, 5 \times 10^{-6}, 3 \times 10^{-5}\}$ and fixed $N = 100$.

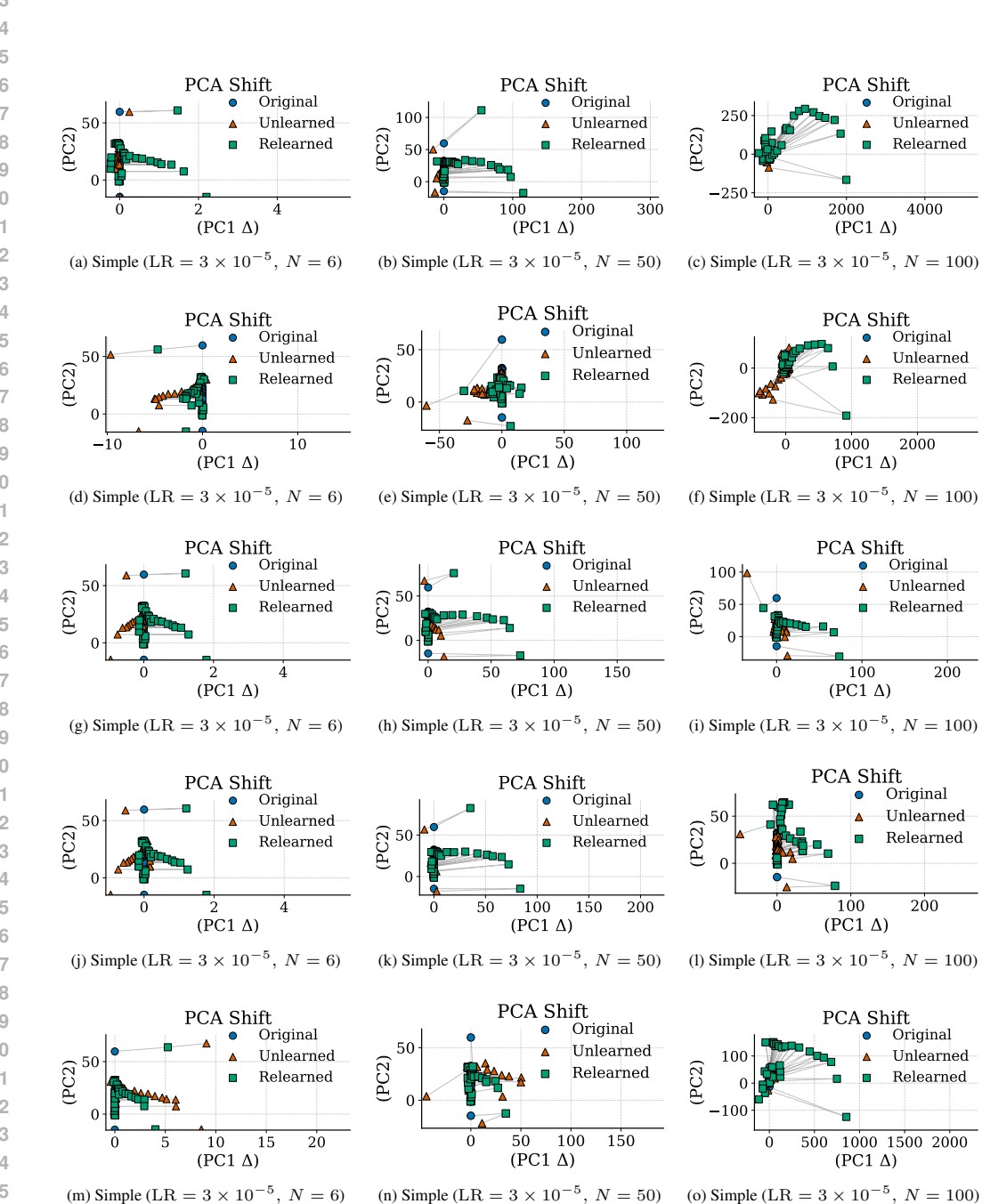

Figure 12: PCA Shift Across Layers. Each row shows results under different unlearning methods: GA+GD (a–c), GA+KL (d–f), NPO (g–i), NPO+KL (j–l), and Rlable (m–o). Simple task on Yi-6B with fixed learning rate $\text{LR} = 3 \times 10^{-5}$ and varying unlearning requests $N \in \{6, 50, 100\}$.

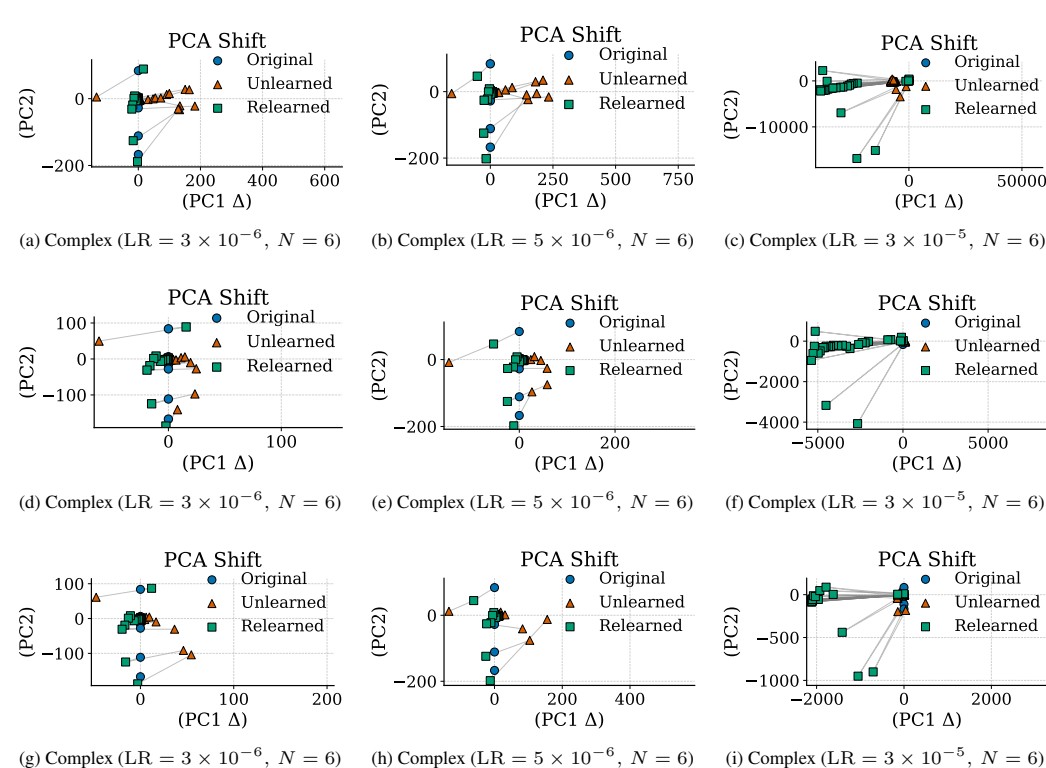

Figure 13: PCA Shift Across Layers. Each row shows results under different unlearning methods: GA (a-c) NPO (d–f), Rlable (g–j). All plots are for the complex task on Qwen2.5-7B, using three learning rates $\{3 \times 10^{-6}, 5 \times 10^{-6}, 3 \times 10^{-5}\}$ and fixed $N = 6$.

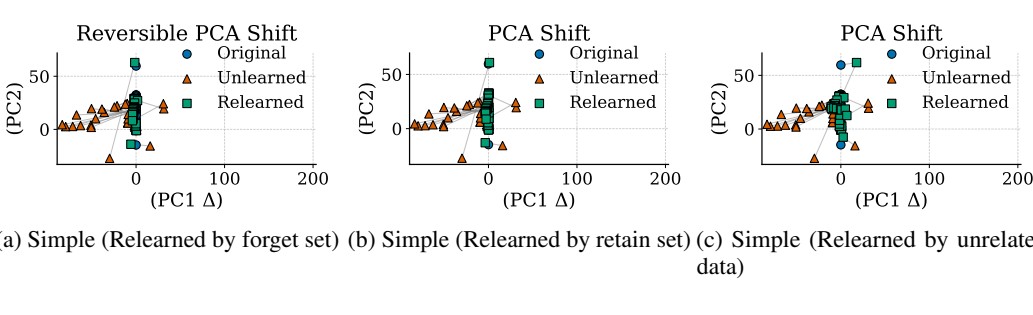

(a) Simple (Relearned by forget set)  (b) Simple (Relearned by retain set)  (c) Simple (Relearned by unrelated data)

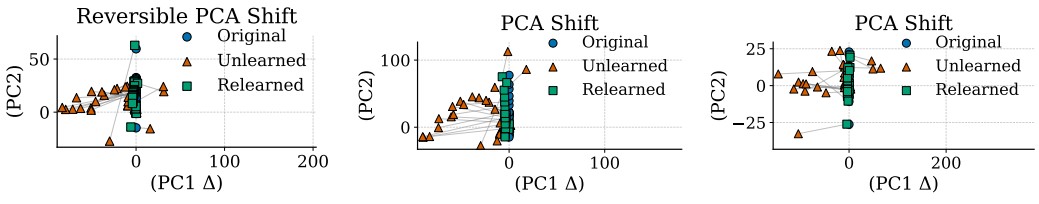

(d) Simple (input data = forget set)  (e) Simple (input data = retain set)  (f) Simple (input data = unrelated data)

Figure 14: PCA Shift Analysis under Varied Relearning and Evaluation Inputs on Yi-6B (Simple Task). (a–c): Relearning is performed using the forget set, retain set, or unrelated data respectively. (d–f): PCA shift is measured using the forget set, retain set, or unrelated data as evaluation input.

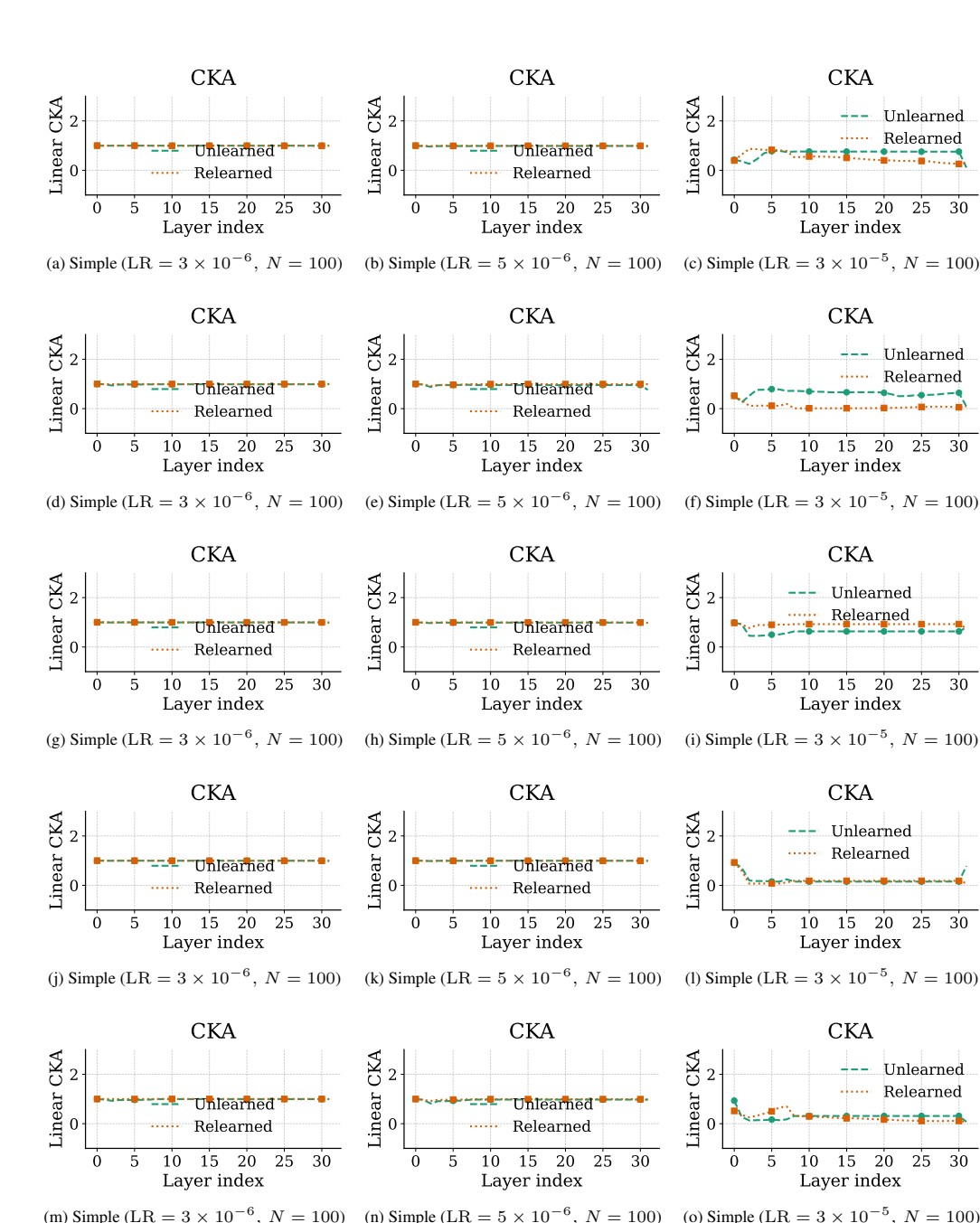

Figure 15: CKA Across Layers. Each row shows results under different unlearning methods: GA+GD (a–c), GA+KL (d–f), NPO (g–i), NPO+KL (j–l), and Rlable (m–o). All plots are for the simple task on Yi-6B, using three learning rates $\{3 \times 10^{-6}, 5 \times 10^{-6}, 3 \times 10^{-5}\}$ and fixed $N = 100$.

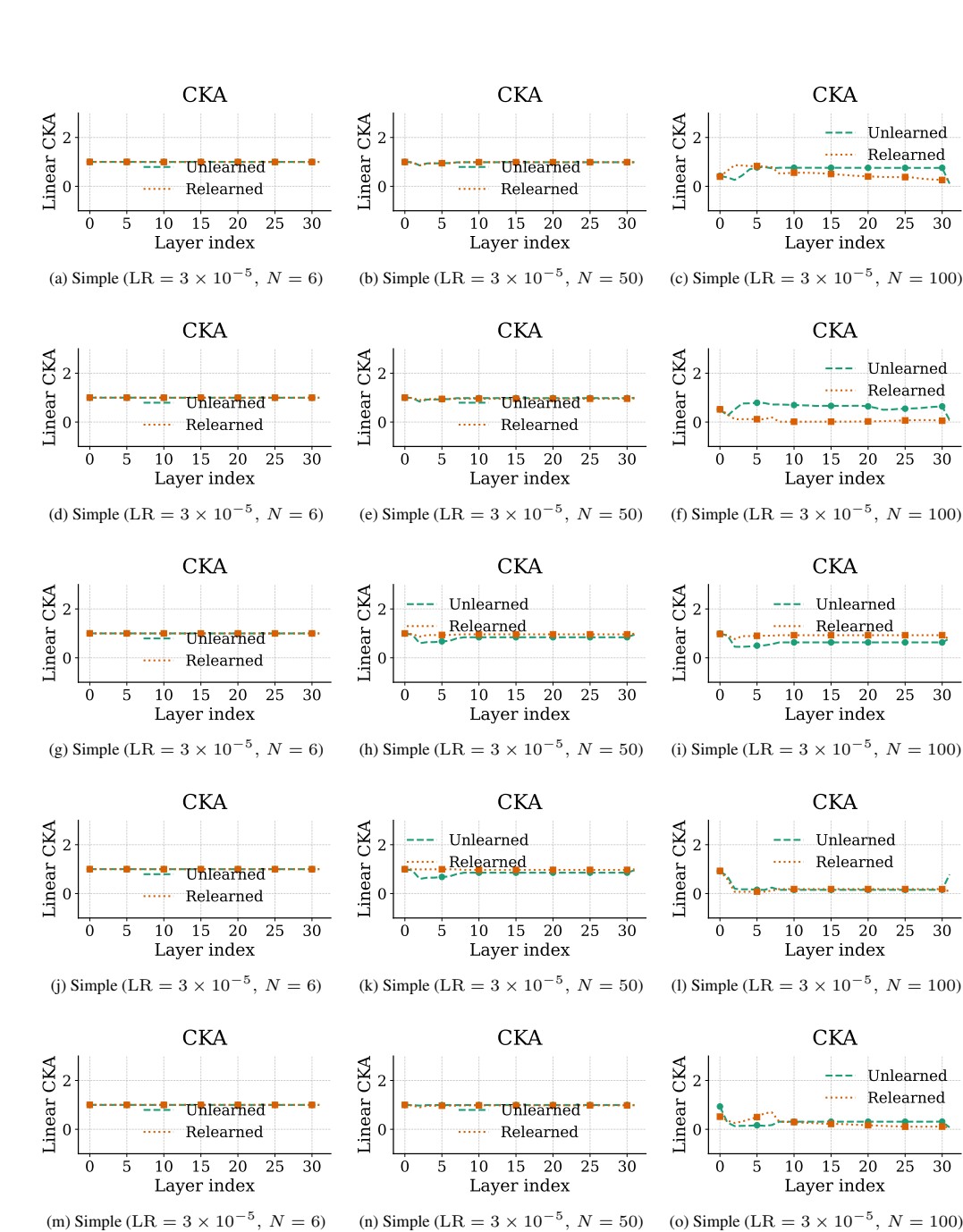

Figure 16: CKA Across Layers. Each row shows results under different unlearning methods: GA+GD (a–c), GA+KL (d–f), NPO (g–i), NPO+KL (j–l), and Rlable (m–o). Simple task on Yi-6B with fixed learning rate LR = $3 \times 10^{-5}$ and varying unlearning requests $N \in \{6, 50, 100\}$.

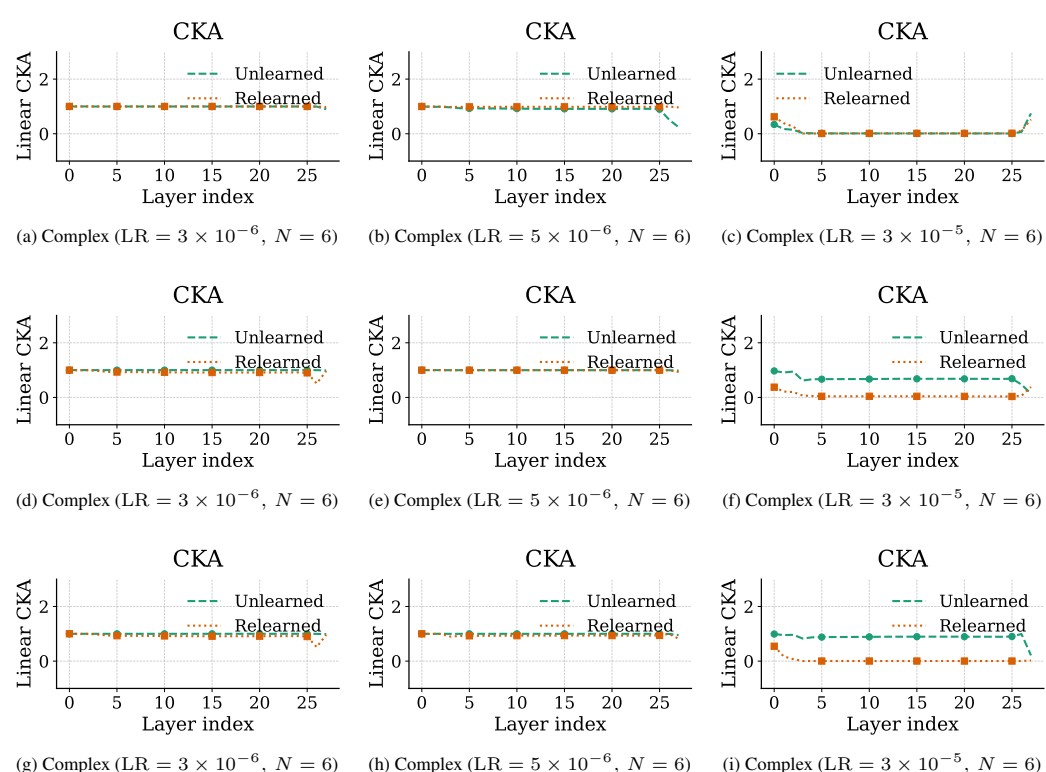

Figure 17: CKA Across Layers. Each row shows results under different unlearning methods: GA (a-c) NPO (d–f), Rlable (g–j). All plots are for the complex task on Qwen2.5-7B, using three learning rates $\{3 \times 10^{-6}, 5 \times 10^{-6}, 3 \times 10^{-5}\}$ and fixed $N = 6$.

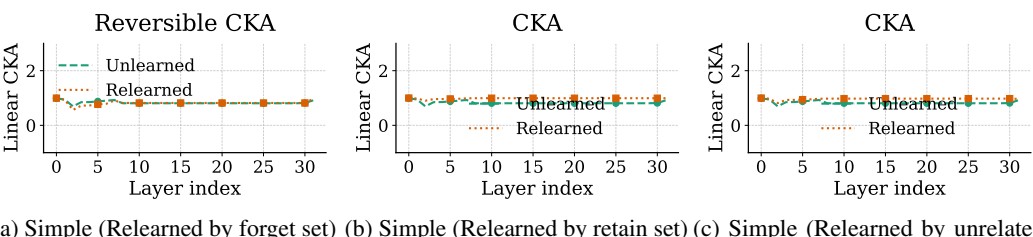

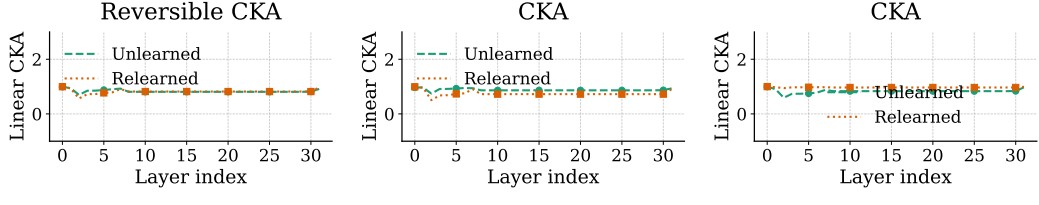

Figure 18: CKA Analysis under Varied Relearning and Evaluation Inputs on Yi-6B (Simple Task). (a–c): Relearning is performed using the forget set, retain set, or unrelated data respectively. (d–f): CKA is measured using the forget set, retain set, or unrelated data as evaluation input.

(a) Simple LR=$3 \times 10^{-6}$, Layer 28  (b) Simple LR=$5 \times 10^{-6}$, Layer 28  (c) Simple LR=$3 \times 10^{-5}$, Layer 28

(d) Simple LR=$3 \times 10^{-6}$, Layer 22  (e) Simple LR=$5 \times 10^{-6}$, Layer 22  (f) Simple LR=$3 \times 10^{-5}$, Layer 22

(g) Simple LR=$3 \times 10^{-6}$, Layer 13  (h) Simple LR=$5 \times 10^{-6}$, Layer 13  (i) Simple LR=$3 \times 10^{-5}$, Layer 13

(j) Simple LR=$3 \times 10^{-6}$, Layer 4  (k) Simple LR=$5 \times 10^{-6}$, Layer 4  (l) Simple LR=$3 \times 10^{-5}$, Layer 4

(m) Simple LR=$3 \times 10^{-6}$, Layer 1  (n) Simple LR=$5 \times 10^{-6}$, Layer 1  (o) Simple LR=$3 \times 10^{-5}$, Layer 1

Figure 19: FIM for GA Across Layers. All plots are for the simple task on Yi-6B, using three learning rates $\{3 \times 10^{-6}, 5 \times 10^{-6}, 3 \times 10^{-5}\}$ and fixed $N = 100$.

(a) Simple LR=$3 \times 10^{-6}$, Layer 31  (b) Simple LR=$5 \times 10^{-6}$, Layer 31  (c) Simple LR=$3 \times 10^{-5}$, Layer 31

(d) Simple LR=$3 \times 10^{-6}$, Layer 28  (e) Simple LR=$5 \times 10^{-6}$, Layer 28  (f) Simple LR=$3 \times 10^{-5}$, Layer 28

(g) Simple LR=$3 \times 10^{-6}$, Layer 22  (h) Simple LR=$5 \times 10^{-6}$, Layer 22  (i) Simple LR=$3 \times 10^{-5}$, Layer 22

(j) Simple LR=$3 \times 10^{-6}$, Layer 13  (k) Simple LR=$5 \times 10^{-6}$, Layer 13  (l) Simple LR=$3 \times 10^{-5}$, Layer 13

(m) Simple LR=$3 \times 10^{-6}$, Layer 4  (n) Simple LR=$5 \times 10^{-6}$, Layer 4  (o) Simple LR=$3 \times 10^{-5}$, Layer 4

(p) Simple LR=$3 \times 10^{-6}$, Layer 1  (q) Simple LR=$5 \times 10^{-6}$, Layer 1  (r) Simple LR=$3 \times 10^{-5}$, Layer 1

Figure 20: FIM for GA+GD Across Layers. All plots are for the simple task on Yi-6B, using three learning rates $\{3 \times 10^{-6}, 5 \times 10^{-6}, 3 \times 10^{-5}\}$ and fixed $N = 100$.

(a) Simple LR=$3 \times 10^{-6}$, Layer 31 (b) Simple LR=$5 \times 10^{-6}$, Layer 31 (c) Simple LR=$3 \times 10^{-5}$, Layer 31

(d) Simple LR=$3 \times 10^{-6}$, Layer 28 (e) Simple LR=$5 \times 10^{-6}$, Layer 28 (f) Simple LR=$3 \times 10^{-5}$, Layer 28

(g) Simple LR=$3 \times 10^{-6}$, Layer 22 (h) Simple LR=$5 \times 10^{-6}$, Layer 22 (i) Simple LR=$3 \times 10^{-5}$, Layer 22

(j) Simple LR=$3 \times 10^{-6}$, Layer 13 (k) Simple LR=$5 \times 10^{-6}$, Layer 13 (l) Simple LR=$3 \times 10^{-5}$, Layer 13

(m) Simple LR=$3 \times 10^{-6}$, Layer 4 (n) Simple LR=$5 \times 10^{-6}$, Layer 4 (o) Simple LR=$3 \times 10^{-5}$, Layer 4

(p) Simple LR=$3 \times 10^{-6}$, Layer 1 (q) Simple LR=$5 \times 10^{-6}$, Layer 1 (r) Simple LR=$3 \times 10^{-5}$, Layer 1

Figure 21: FIM for GA+KL Across Layers. All plots are for the simple task on Yi-6B, using three learning rates $\{3 \times 10^{-6}, 5 \times 10^{-6}, 3 \times 10^{-5}\}$ and fixed $N = 100$.

(a) Simple LR=$3 \times 10^{-6}$, Layer 31    (b) Simple LR=$5 \times 10^{-6}$, Layer 31    (c) Simple LR=$3 \times 10^{-5}$, Layer 31

(d) Simple LR=$3 \times 10^{-6}$, Layer 28    (e) Simple LR=$5 \times 10^{-6}$, Layer 28    (f) Simple LR=$3 \times 10^{-5}$, Layer 28

(g) Simple LR=$3 \times 10^{-6}$, Layer 22    (h) Simple LR=$5 \times 10^{-6}$, Layer 22    (i) Simple LR=$3 \times 10^{-5}$, Layer 22

(j) Simple LR=$3 \times 10^{-6}$, Layer 13    (k) Simple LR=$5 \times 10^{-6}$, Layer 13    (l) Simple LR=$3 \times 10^{-5}$, Layer 13

(m) Simple LR=$3 \times 10^{-6}$, Layer 4    (n) Simple LR=$5 \times 10^{-6}$, Layer 4    (o) Simple LR=$3 \times 10^{-5}$, Layer 4

(p) Simple LR=$3 \times 10^{-6}$, Layer 1    (q) Simple LR=$5 \times 10^{-6}$, Layer 1    (r) Simple LR=$3 \times 10^{-5}$, Layer 1

Figure 22: FIM for NPO Across Layers. All plots are for the simple task on Yi-6B, using three learning rates $\{3 \times 10^{-6}, 5 \times 10^{-6}, 3 \times 10^{-5}\}$ and fixed $N = 100$.

(a) Simple LR=$3 \times 10^{-6}$, Layer 31  (b) Simple LR=$5 \times 10^{-6}$, Layer 31  (c) Simple LR=$3 \times 10^{-5}$, Layer 31

(d) Simple LR=$3 \times 10^{-6}$, Layer 28  (e) Simple LR=$5 \times 10^{-6}$, Layer 28  (f) Simple LR=$3 \times 10^{-5}$, Layer 28

(g) Simple LR=$3 \times 10^{-6}$, Layer 22  (h) Simple LR=$5 \times 10^{-6}$, Layer 22  (i) Simple LR=$3 \times 10^{-5}$, Layer 22

(j) Simple LR=$3 \times 10^{-6}$, Layer 13  (k) Simple LR=$5 \times 10^{-6}$, Layer 13  (l) Simple LR=$3 \times 10^{-5}$, Layer 13

(m) Simple LR=$3 \times 10^{-6}$, Layer 4  (n) Simple LR=$5 \times 10^{-6}$, Layer 4  (o) Simple LR=$3 \times 10^{-5}$, Layer 4

(p) Simple LR=$3 \times 10^{-6}$, Layer 1  (q) Simple LR=$5 \times 10^{-6}$, Layer 1  (r) Simple LR=$3 \times 10^{-5}$, Layer 1

Figure 23: FIM for NPO+KL Across Layers. All plots are for the simple task on Yi-6B, using three learning rates $\{3 \times 10^{-6}, 5 \times 10^{-6}, 3 \times 10^{-5}\}$ and fixed $N = 100$.

(a) Simple LR=$3 \times 10^{-6}$, Layer 31  (b) Simple LR=$5 \times 10^{-6}$, Layer 31  (c) Simple LR=$3 \times 10^{-5}$, Layer 31

(d) Simple LR=$3 \times 10^{-6}$, Layer 28  (e) Simple LR=$5 \times 10^{-6}$, Layer 28  (f) Simple LR=$3 \times 10^{-5}$, Layer 28

(g) Simple LR=$3 \times 10^{-6}$, Layer 22  (h) Simple LR=$5 \times 10^{-6}$, Layer 22  (i) Simple LR=$3 \times 10^{-5}$, Layer 22

(j) Simple LR=$3 \times 10^{-6}$, Layer 13  (k) Simple LR=$5 \times 10^{-6}$, Layer 13  (l) Simple LR=$3 \times 10^{-5}$, Layer 13

(m) Simple LR=$3 \times 10^{-6}$, Layer 4  (n) Simple LR=$5 \times 10^{-6}$, Layer 4  (o) Simple LR=$3 \times 10^{-5}$, Layer 4

(p) Simple LR=$3 \times 10^{-6}$, Layer 1  (q) Simple LR=$5 \times 10^{-6}$, Layer 1  (r) Simple LR=$3 \times 10^{-5}$, Layer 1

Figure 24: FIM for Rlable Across Layers. All plots are for the simple task on Yi-6B, using three learning rates $\{3 \times 10^{-6}, 5 \times 10^{-6}, 3 \times 10^{-5}\}$ and fixed $N = 100$.

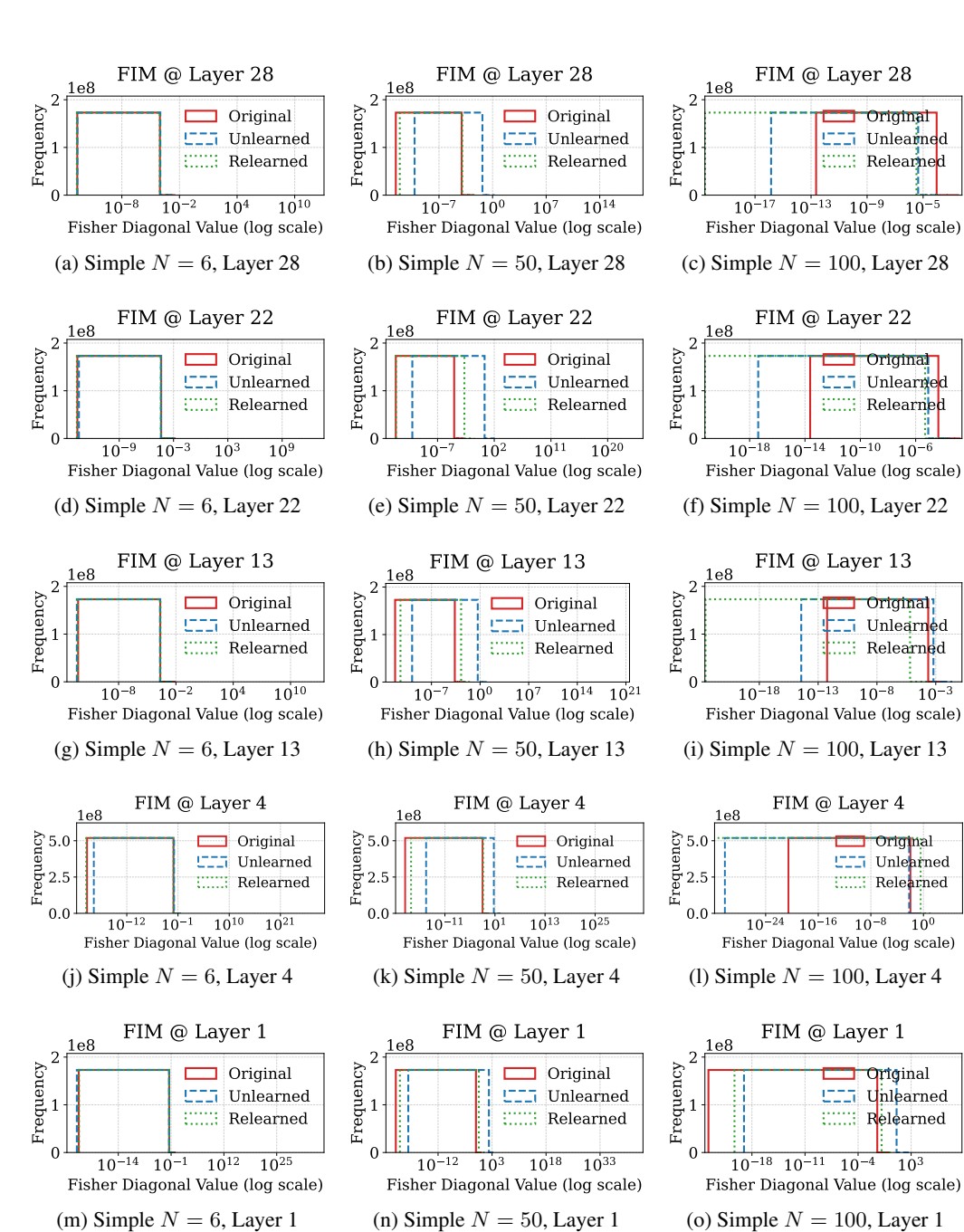

Figure 25: FIM for GA Across Layers. Simple task on Yi-6B with fixed learning rate LR $= 3 \times 10^{-5}$ and varying unlearning requests $N \in \{6, 50, 100\}$.

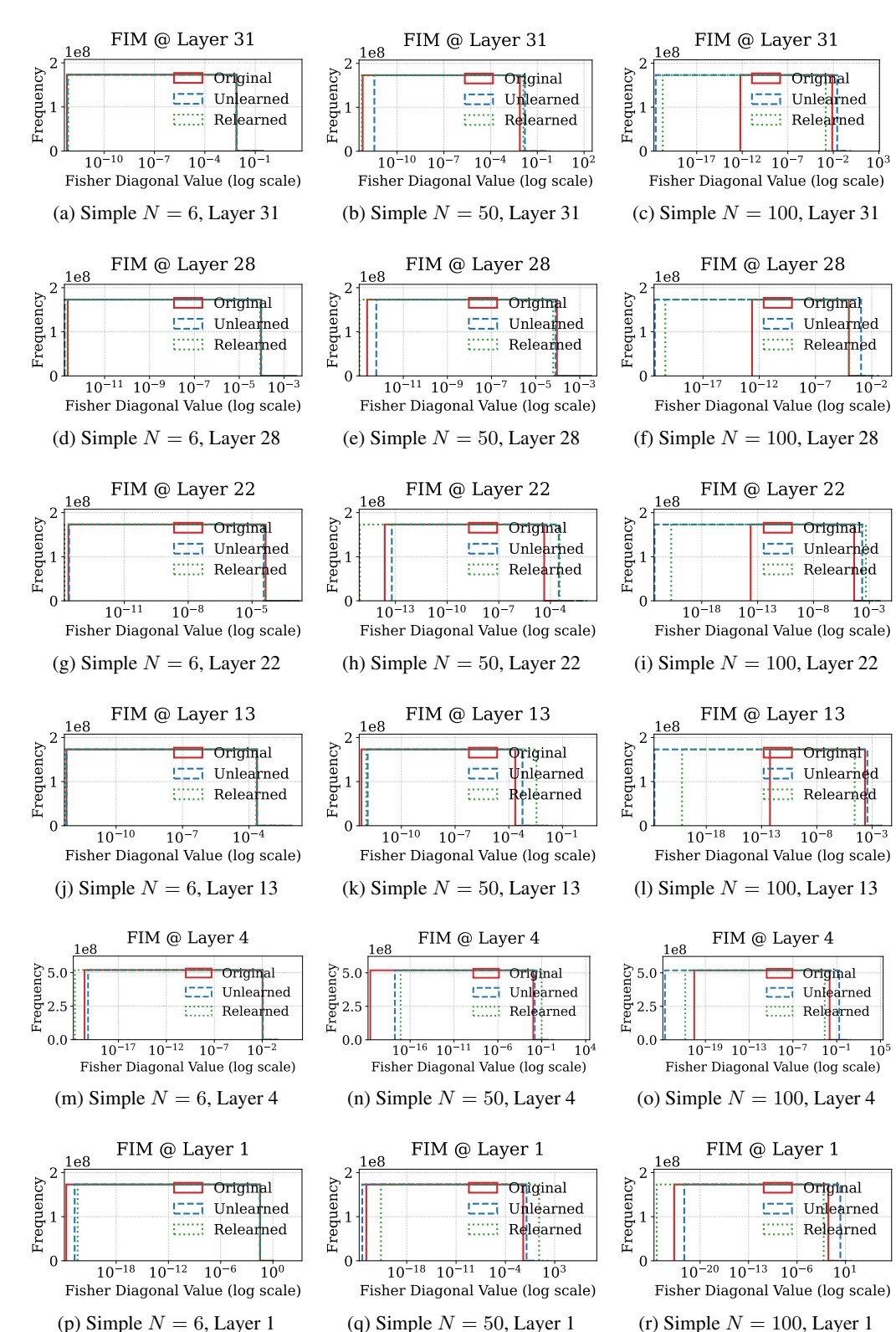

Figure 26: FIM for GA+GD Across Layers. Simple task on Yi-6B with fixed learning rate LR = $3 \times 10^{-5}$ and varying unlearning requests $N \in \{6, 50, 100\}$.

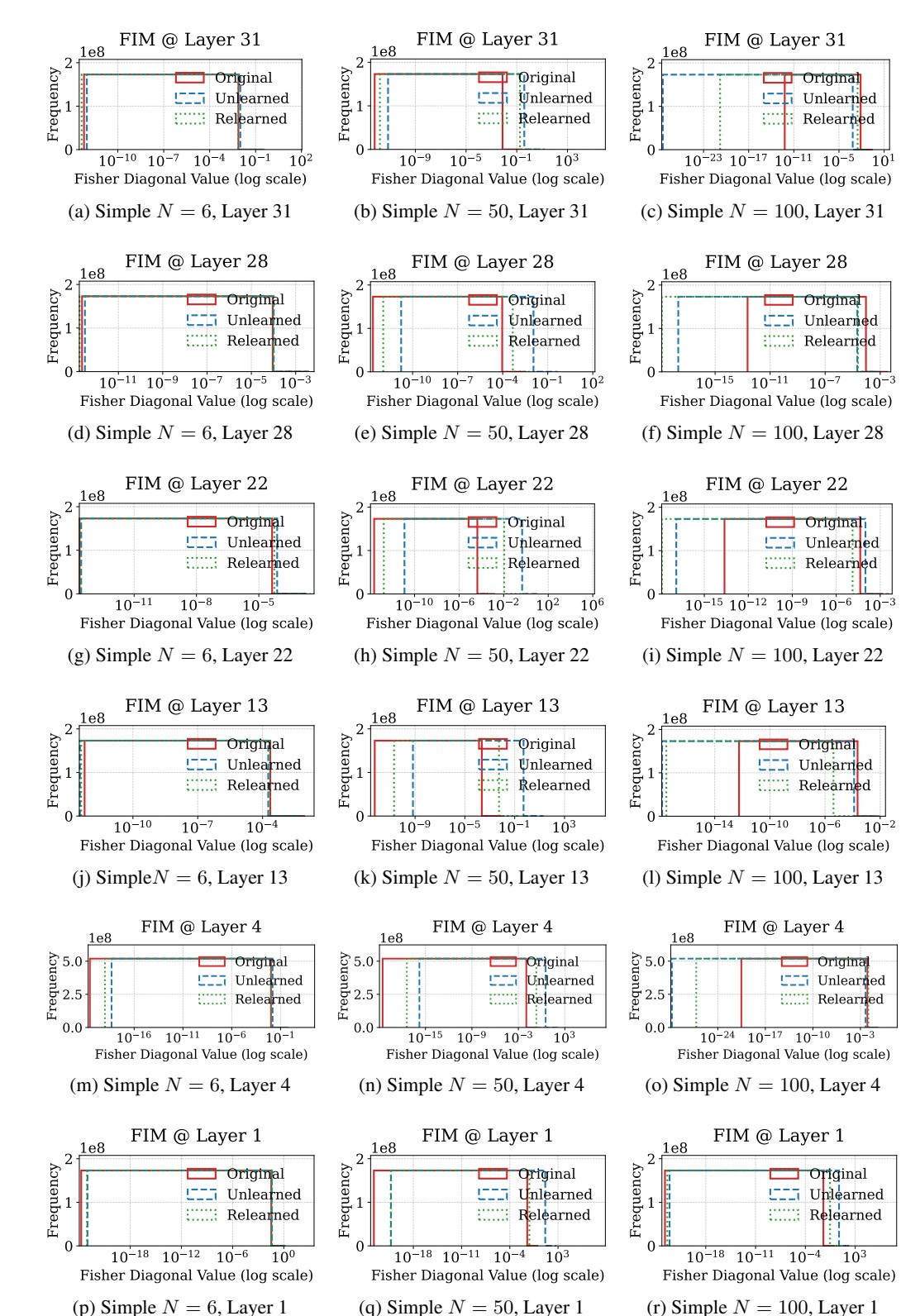

Figure 27: FIM for GA+KL Across Layers. Simple task on Yi-6B with fixed learning rate $\text{LR} = 3 \times 10^{-5}$ and varying unlearning requests $N \in \{6, 50, 100\}$.

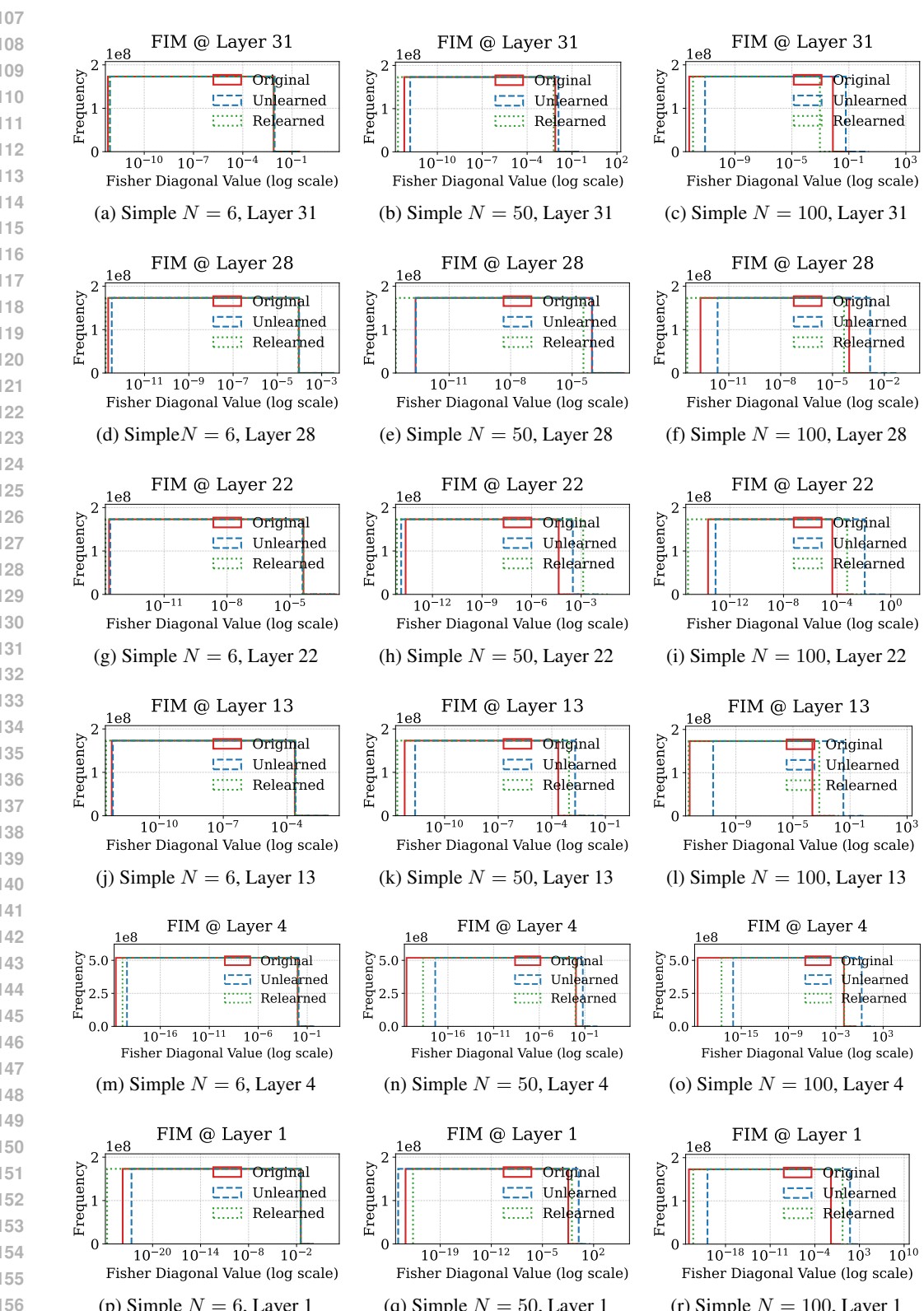

Figure 28: FIM for NPO Across Layers. Simple task on Yi-6B with fixed learning rate LR $= 3 \times 10^{-5}$ and varying unlearning requests $N \in \{6, 50, 100\}$.

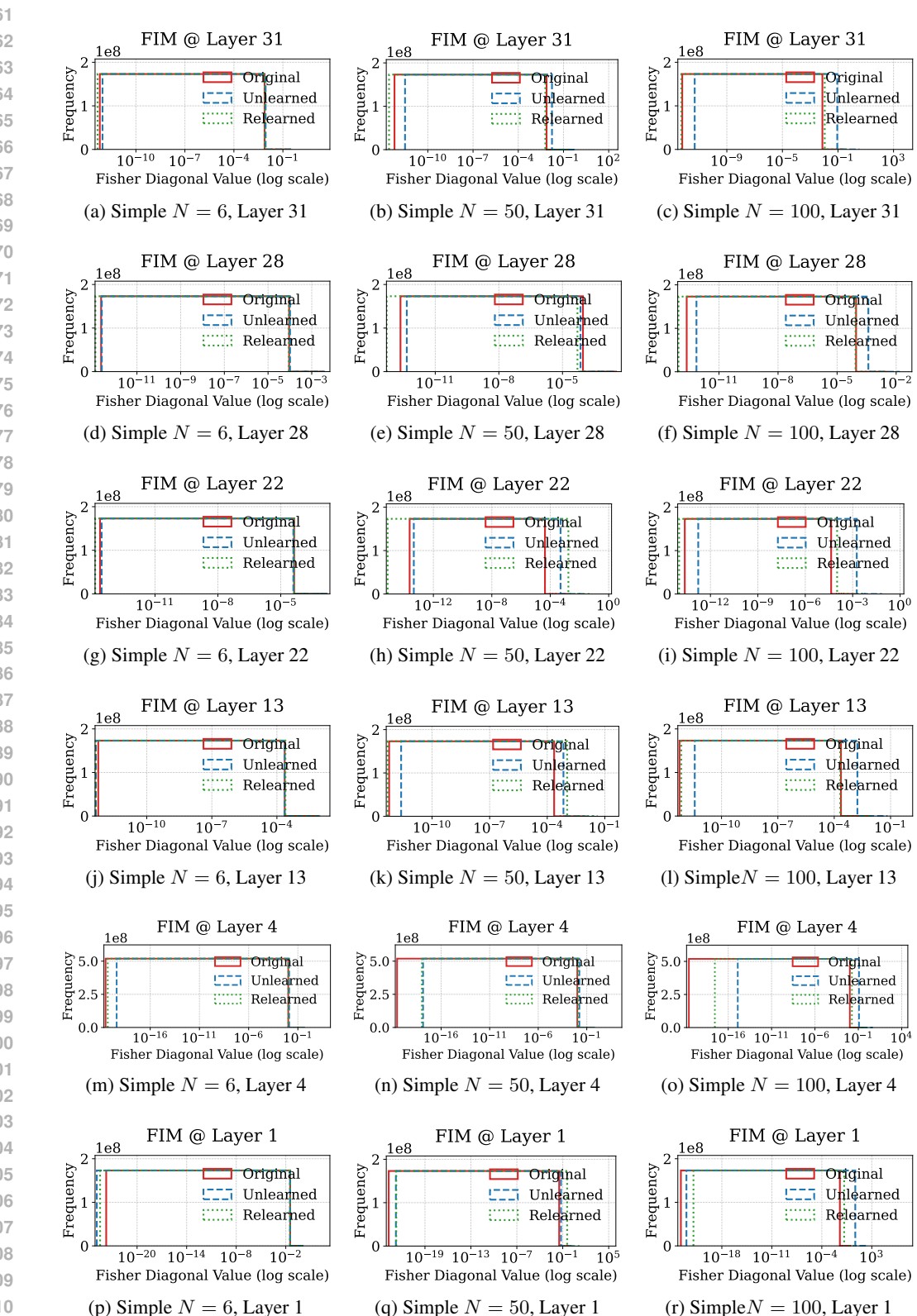

Figure 29: FIM for NPO+KL Across Layers. Simple task on Yi-6B with fixed learning rate LR = $3 \times 10^{-5}$ and varying unlearning requests $N \in \{6, 50, 100\}$.

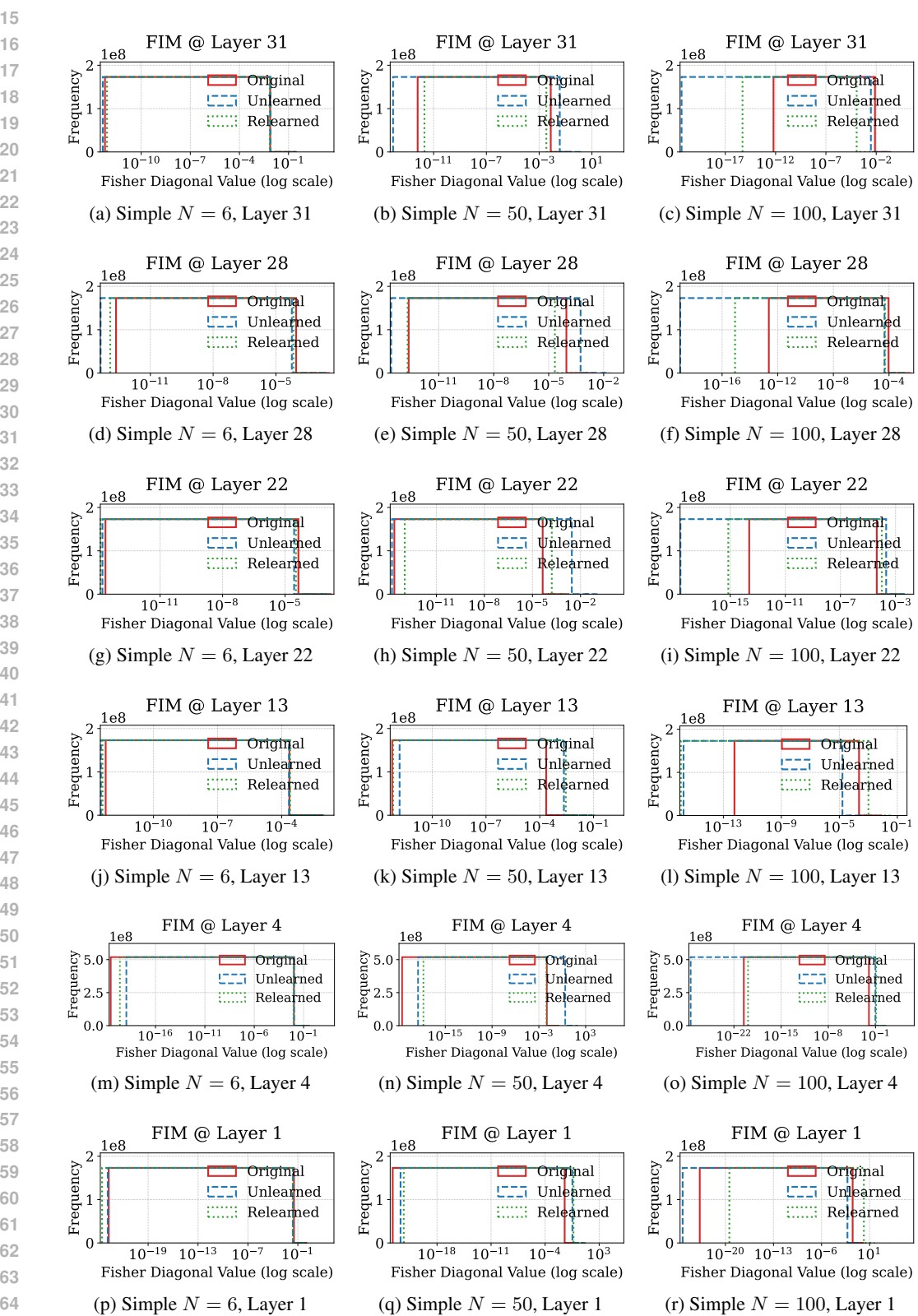

Figure 30: FIM for Rlable Across Layers. Simple task on Yi-6B with fixed learning rate LR $= 3 \times 10^{-5}$ and varying unlearning requests $N \in \{6, 50, 100\}$.

(a) Complex LR=$3 \times 10^{-6}$, Layer 28(b) Complex LR=$5 \times 10^{-6}$, Layer 28(c) Complex LR=$3 \times 10^{-5}$, Layer 28

(d) Complex LR=$3 \times 10^{-6}$, Layer 24(e) Complex LR=$5 \times 10^{-6}$, Layer 24(f) Complex LR=$3 \times 10^{-5}$, Layer 24

(g) Complex LR=$3 \times 10^{-6}$, Layer 12(h) Complex LR=$5 \times 10^{-6}$, Layer 12(i) Complex LR=$3 \times 10^{-5}$, Layer 12

(j) Complex LR=$3 \times 10^{-6}$, Layer 4 (k) Complex LR=$5 \times 10^{-6}$, Layer 4 (l) Complex LR=$3 \times 10^{-5}$, Layer 4

(m) Complex LR=$3 \times 10^{-6}$, Layer 1(n) Complex LR=$5 \times 10^{-6}$, Layer 1(o) Complex LR=$3 \times 10^{-5}$, Layer 1

Figure 31: FIM for GA Across Layers. All plots are for the complex task on Qwen2.5-7B, using three learning rates $\{3 \times 10^{-6}, 5 \times 10^{-6}, 3 \times 10^{-5}\}$ and fixed $N = 6$.

(a) Complex LR=$3 \times 10^{-6}$, Layer 28 (b) Complex LR=$5 \times 10^{-6}$, Layer 28 (c) Complex LR=$3 \times 10^{-5}$, Layer 28

(d) Complex LR=$3 \times 10^{-6}$, Layer 24 (e) Complex LR=$5 \times 10^{-6}$, Layer 24 (f) Complex LR=$3 \times 10^{-5}$, Layer 24

(g) Complex LR=$3 \times 10^{-6}$, Layer 12 (h) Complex LR=$5 \times 10^{-6}$, Layer 12 (i) Complex LR=$3 \times 10^{-5}$, Layer 12

(j) Complex LR=$3 \times 10^{-6}$, Layer 4 (k) Complex LR=$5 \times 10^{-6}$, Layer 4 (l) Complex LR=$3 \times 10^{-5}$, Layer 4

(m) Complex LR=$3 \times 10^{-6}$, Layer 1 (n) Complex LR=$5 \times 10^{-6}$, Layer 1 (o) Complex LR=$3 \times 10^{-5}$, Layer 1

Figure 32: FIM for NPO Across Layers. All plots are for the complex task on Qwen2.5-7B, using three learning rates $\{3 \times 10^{-6}, 5 \times 10^{-6}, 3 \times 10^{-5}\}$ and fixed $N = 6$.

(a) Complex LR=$3 \times 10^{-6}$, Layer 28 (b) Complex LR=$5 \times 10^{-6}$, Layer 28 (c) Complex LR=$3 \times 10^{-5}$, Layer 28

(d) Complex LR=$3 \times 10^{-6}$, Layer 24 (e) Complex LR=$5 \times 10^{-6}$, Layer 24 (f) Complex LR=$3 \times 10^{-5}$, Layer 24

(g) Complex LR=$3 \times 10^{-6}$, Layer 12 (h) Complex LR=$5 \times 10^{-6}$, Layer 12 (i) Complex LR=$3 \times 10^{-5}$, Layer 12

(j) Complex LR=$3 \times 10^{-6}$, Layer 4 (k) Complex LR=$5 \times 10^{-6}$, Layer 4 (l) Complex LR=$3 \times 10^{-5}$, Layer 4

(m) Complex LR=$3 \times 10^{-6}$, Layer 1 (n) Complex LR=$5 \times 10^{-6}$, Layer 1 (o) Complex LR=$3 \times 10^{-5}$, Layer 1

Figure 33: FIM for Rlable Across Layers. All plots are for the complex task on Qwen2.5-7B, using three learning rates $\{3 \times 10^{-6}, 5 \times 10^{-6}, 3 \times 10^{-5}\}$ and fixed $N = 6$.

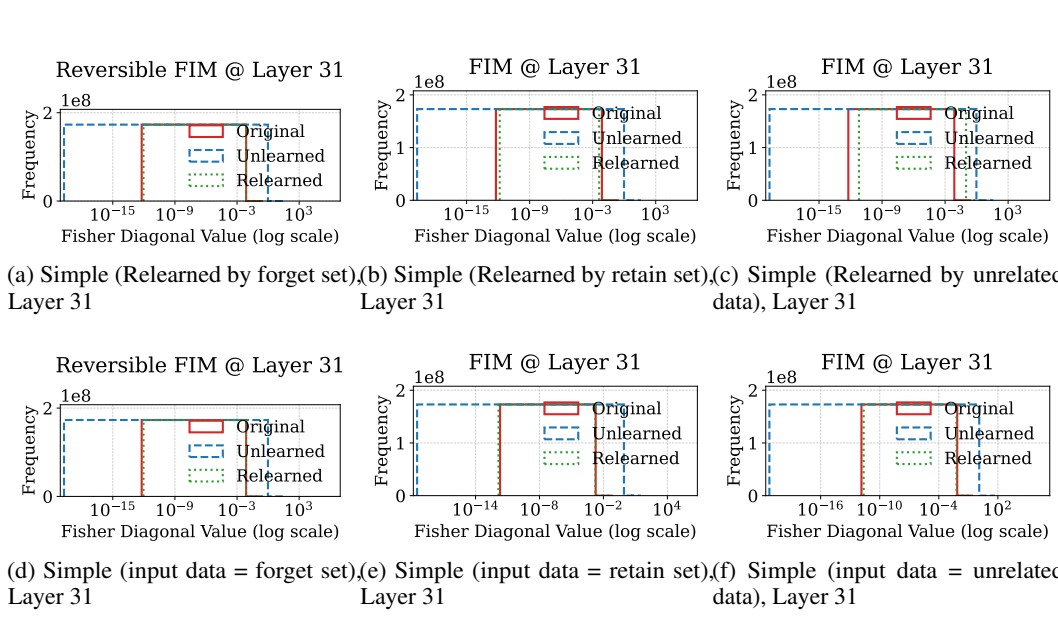

(a) Simple (Relearned by forget set), Layer 31

(b) Simple (Relearned by retain set), Layer 31

(c) Simple (Relearned by unrelated data), Layer 31

(d) Simple (input data = forget set), Layer 31

(e) Simple (input data = retain set), Layer 31

(f) Simple (input data = unrelated data), Layer 31

Figure 34: FIM in layer 31 under Varied Relearning and Evaluation Inputs on Yi-6B (Simple Task). (a–c): Relearning is performed using the forget set, retain set, or unrelated data respectively. (d–f): FIM is measured using the forget set, retain set, or unrelated data as evaluation input.

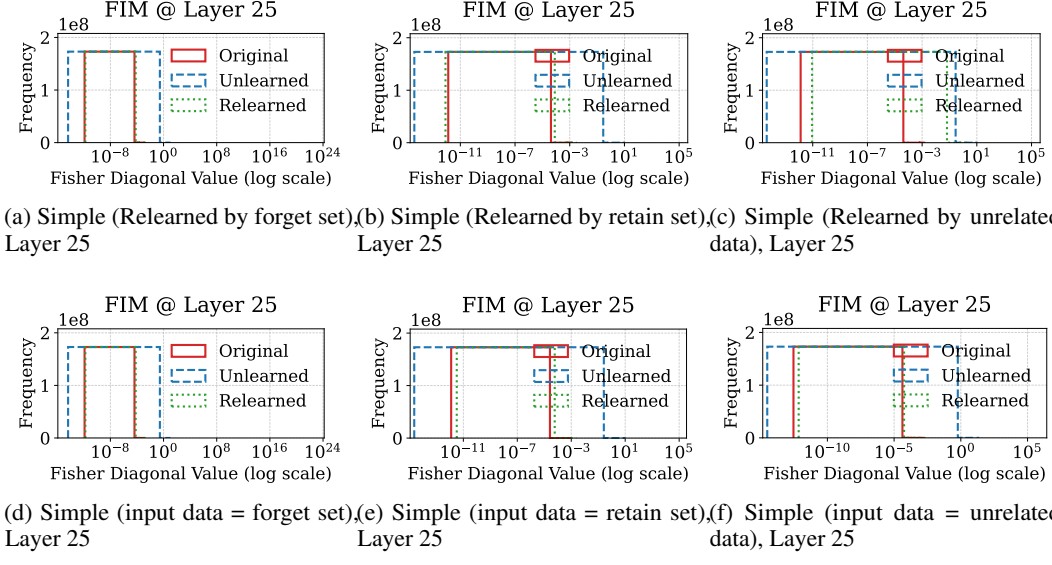

(a) Simple (Relearned by forget set), Layer 25

(b) Simple (Relearned by retain set), Layer 25

(c) Simple (Relearned by unrelated data), Layer 25

(d) Simple (input data = forget set), Layer 25

(e) Simple (input data = retain set), Layer 25

(f) Simple (input data = unrelated data), Layer 25

Figure 35: FIM in layer 25 under Varied Relearning and Evaluation Inputs on Yi-6B (Simple Task). (a–c): Relearning is performed using the forget set, retain set, or unrelated data respectively. (d–f): FIM is measured using the forget set, retain set, or unrelated data as evaluation input.

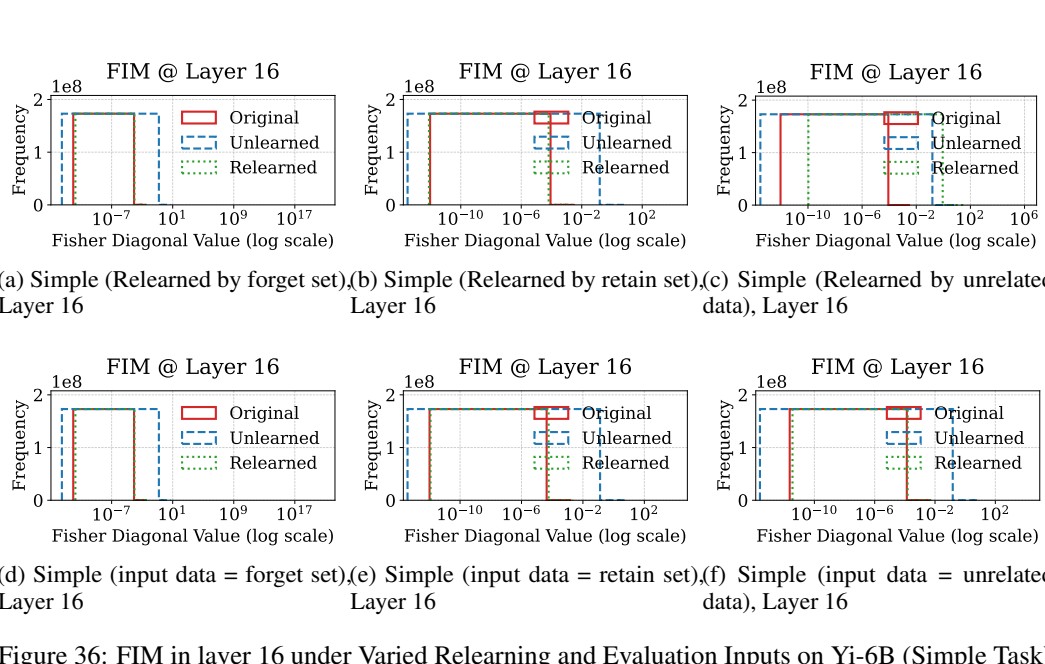

(a) Simple (Relearned by forget set), Layer 16
(b) Simple (Relearned by retain set), Layer 16
(c) Simple (Relearned by unrelated data), Layer 16

(d) Simple (input data = forget set), Layer 16
(e) Simple (input data = retain set), Layer 16
(f) Simple (input data = unrelated data), Layer 16

Figure 36: FIM in layer 16 under Varied Relearning and Evaluation Inputs on Yi-6B (Simple Task). (a–c): Relearning is performed using the forget set, retain set, or unrelated data respectively. (d–f): FIM is measured using the forget set, retain set, or unrelated data as evaluation input.

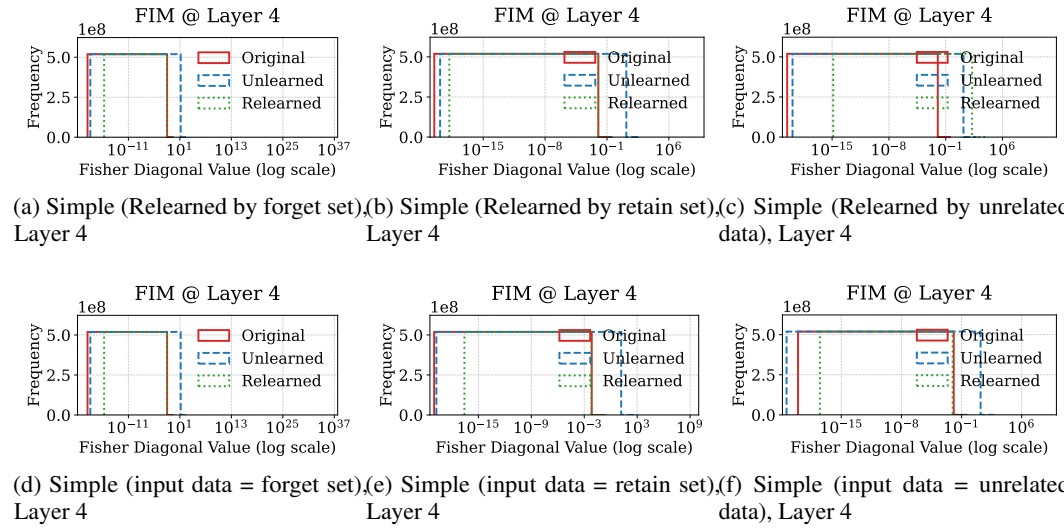

(a) Simple (Relearned by forget set), Layer 4
(b) Simple (Relearned by retain set), Layer 4
(c) Simple (Relearned by unrelated data), Layer 4

(d) Simple (input data = forget set), Layer 4
(e) Simple (input data = retain set), Layer 4
(f) Simple (input data = unrelated data), Layer 4

Figure 37: FIM in layer 4 under Varied Relearning and Evaluation Inputs on Yi-6B (Simple Task). (a–c): Relearning is performed using the forget set, retain set, or unrelated data respectively. (d–f): FIM is measured using the forget set, retain set, or unrelated data as evaluation input.

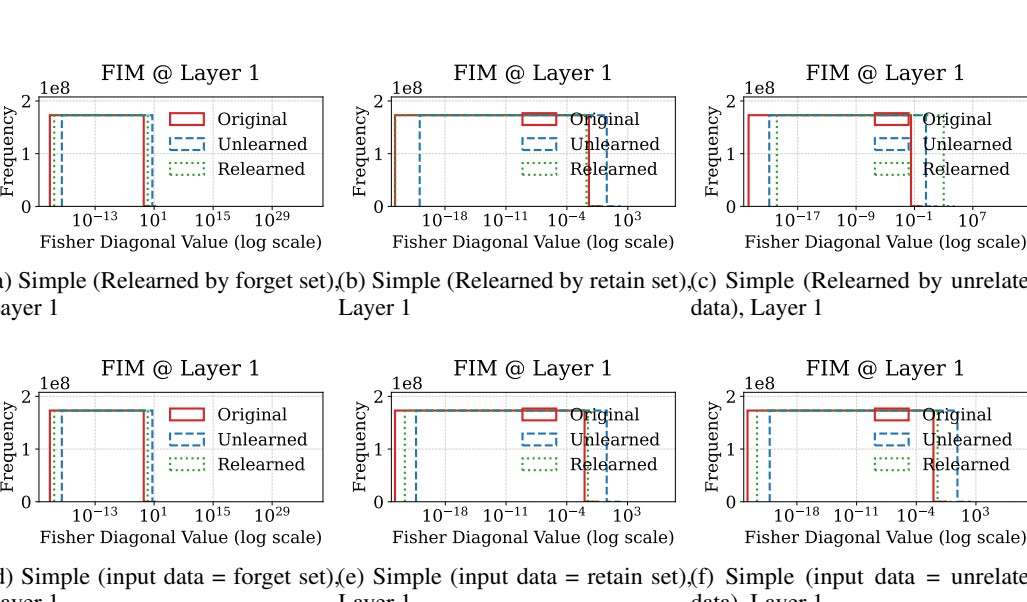

(a) Simple (Relearned by forget set), Layer 1

(b) Simple (Relearned by retain set), Layer 1

(c) Simple (Relearned by unrelated data), Layer 1

(d) Simple (input data = forget set), Layer 1

(e) Simple (input data = retain set), Layer 1

(f) Simple (input data = unrelated data), Layer 1

Figure 38: FIM in layer 1 under Varied Relearning and Evaluation Inputs on Yi-6B (Simple Task). (a–c): Relearning is performed using the forget set, retain set, or unrelated data respectively. (d–f): FIM is measured using the forget set, retain set, or unrelated data as evaluation input.

