# OpenReview forum: "Unlearning Isn't Deletion: Investigating Reversibility of Machine Unlearning in LLMs"
_ICLR.cc/2026/Conference — Submitted to ICLR 2026_

### Official Review · Reviewer_n9vs · 2025-10-25

**Soundness:** 2
**Presentation:** 3
**Contribution:** 2
**Rating:** 2
**Confidence:** 4

**Summary:**

* This paper evaluates unlearning in two settings: single (where all unlearning requests are available simultaneously) and continual (where unlearning requests arrive sequentially).
* The paper defines four unlearning regimes that vary along two axes: reversible vs. irreversible (i.e., whether the unlearned knowledge can be recovered by lightweight retraining) and catastrophic vs. non-catastrophic (i.e., whether the unlearning process significantly affects unrelated knowledge).
* Evaluating 3 unlearning methods and 2 LLMs, the paper first shows that unlearning methods cannot remove knowledge irreversibly.
* Then, the paper describes and applies several methods for measuring representational similarity between the original LLM and its unlearned and retrained variants, finding that representational similarities correlate with reversibility of unlearning.
* This is complemented by a theoretical analysis and a short case study of achieving irreversible, non-catastrophic unlearning.
* The paper concludes that achieving irreversible unlearning remains hard, and representation analysis offers a new perspective on unlearning beyond accuracy on retain and forget sets

**Strengths:**

**(S1)** The paper successfully demonstrates that current methods do not achieve irreversible and non-catastrophic unlearning

**(S2)** The introduced taxonomy may be helpful in future work to better systematize and discuss the achievements of new unlearning methods

**(S3)** The paper makes a convincing argument that accuracy metrics alone give an insufficient impression of unlearning success, and analyzing the model's internal representations can give important insights beyond accuracy

**(S4)** The paper contains one example of successful irreversible and non-catastrophic unlearning, demonstrating that this goal may be achievable.

**Weaknesses:**

**(W1)** One major concern is originality: The evaluation of unlearning methods successfully confirms that current methods do not achieve irreversible unlearning, but this is a known fact, as the paper also mentions (e.g., [24]). Likewise, the observation that models break down when applying multiple edits in continual learning has been reported before, e.g. [a]. Finally, none of the proposed metrics for representation analysis is novel.

**(W2)** The paper does not contain any actionable insights. The representation analysis confirms that larger representational dissimilarity correlates with irreversible unlearning, but this is expected as the original model becomes harder to reconstruct the further the parameters move away from it. More importantly, the paper does not give practical tools, for example, how representation similarity can reliably predict successful and irreversible unlearning, which would be very helpful in practice. Overall, the takeaways from this analysis remain unclear: What is the reader to conclude beyond the observation that a larger representation shift correlates with irreversible unlearning?

**(W3)** The theoretical analysis mirrors this problem: It mainly shows how a larger distortion of weights leads to greater dissimilarity (which is intuitive). The connection to irreversible unlearning is not formalized but only claimed in the paragraph starting with line 412. Sec. 5.2 additionally discusses that model outputs are not a reliable indicator of unlearning. However, this is also not a significant finding, as keeping all LLM parameters except those of the last layer frozen while randomizing parameters in the last layer will yield a practically random model (from the black-box perspective), while it is very likely that most information and knowledge learned by the model will continue to be accessible from earlier layers.

**(W4)** The paper overall focuses on a narrow setting where unlearning and catastrophic forgetting are measured through fixed forget and retain sets. However, it does not consider recovering unlearned knowledge through prompt attacks (e.g., [b]) or mechanistic interpretability (e.g., [c], only intended as an example, not available before submission deadline).

**(W5)** LLM unlearning is a popular research area with many methods. Claims that are meant to be generalizable to the entire field, such as the one in this paper, need to be either evaluated on a large set of methods or require a motivation for why the chosen set of unlearning methods is representative and will give such generalizable insights. This aspect can be expanded upon in the current paper.

**(W6)** The experiment in 5.3. appears very interesting, because it directly targets the case of irreversible, non-catastrophic unlearning. This experiment could be one starting to inform more successful unlearning methods. Therefore, I think it would be very interesting to expand this perspective. One concern I have is to what extent this observation is due to the "more constrained relearning conditions" vs. actually successful unlearning.

**(W7)** The supplementary material contains a large number of plots showing the representation similarity measures for different LLM layers. These plots are not individually interpreted or put in context. Their role in the paper is therefore doubtful. If they do not add any tangible value to the paper, consider removing them.

### References
[a] Thede et al.: Understanding the limits of lifelong knowledge editing in llms. In arXiv, 2025\
[b] Patil et al.: Can sensitive information be deleted from llms? objectives for defending against extraction attacks. In ICLR, 2024\
[c] Cywinski et al.: Eliciting Secret Knowledge from Language Models. In arXiv, 2025

**Questions:**

* Which novel perspectives on unlearning does this paper give the community beyond confirming known problems with current methods?
* Which actionable improvements in LLM unlearning are informed by the representation analysis? What are the main insights beyond the expected observation that higher representation dissimilarity correlates with irreversible unlearning?
* How can we motivate the findings in this paper to generalize to most methods for LLM unlearning, even those not evaluated in the paper? How about extraction attacks beyond retraining?

---

> ### Author Response · Authors · 2025-11-14
> **Response to W1, Q1, W2, and Q2**
>
> We thank the reviewer for the constructive comments (which we respond to point by point below) and for recognizing our strengths, including **clarifying the limits of current methods, proposing a useful taxonomy, highlighting the value of representation-level analysis, and demonstrating a case of irreversible non-catastrophic unlearning**.
>
> **#W1&Q1: Clarifying the Originality of Our Framework and Analysis**
>
>
> We agree that prior work has reported isolated findings. For example, recovery after unlearning [1, 2] and breakdown under repeated requests for unlearning [3]. Our contribution is not a new primitive metric, but a unified, representation-level evaluation framework and taxonomy that, to our knowledge, has not been established before for LLM unlearning.
>
> - Unified perspective and taxonomy: We provide the first systematic, representation-level study that jointly analyzes single-request unlearning, continual unlearning, and catastrophic forgetting within one coherent protocol. Using four complementary tools (PCA similarity/shift, CKA, Fisher information) and an operational relearning probe, we delineate four regimes along reversibility and catastrophicity (Table 1), moving beyond the binary distinctions implicit in prior work. We show existing methods almost exclusively occupy reversible regimes, whereas the practically useful target--irreversible, non-catastrophic forgetting--remains unachieved and is distinct from mere model degradation.
>
> - Our controlled relearning protocol treats relearning as an empirical probe of reversibility (not an end in itself), and we demonstrate recovery under forget, retain, and fully unrelated data, revealing structural limits of current methods. We validate regime transitions across models, methods, domains, and hyperparameters, and show that task-level metrics (including MIA AUC) can be unstable and insufficient for assessment.
>
> - While individual diagnostics are known, our novelty lies in their synthesis, the regime taxonomy, and a perturbation-based analysis that explains when localized vs. distributed updates lead to reversible versus irreversible drift. We also introduce a concise summary statistic (mean PCA distance) to quantify layer-wise drift across settings.
>
> To recap, instead of restating known phenomena, we provide a coherent representation-level framework that integrates single and continual unlearning, incorporates catastrophic forgetting into the taxonomy, and explains how and why different regimes emerge, clarifying what would be required to achieve irreversible, non-catastrophic forgetting in practice.
>
> **#W2&Q2: Clarifying the Practical Insights and Takeaways of Our Representation Analysis**
>
> Thank you for the thoughtful comment. Our aim is not merely to restate that "more drift => harder recovery," but to "operationalize" this into a usable, model-agnostic diagnostic tool that practitioners can apply.
>
> Actionable contributions:
> - Unified diagnostic toolkit: PCA similarity/shift, CKA, Fisher information, plus a summary statistic (mean PCA distance) that together provide consistent, layer-resolved signals distinguishing four regimes (reversible/irreversible $\times$ catastrophic/non‑catastrophic). Task-level metrics alone cannot make these distinctions.
>
> - Predictive use within a fixed protocol: Under a specified relearning budget and data source, layer-wise PCA shift and mean PCA distance reliably predict recovery failure (irreversibility), while high CKA and concentrated Fisher spectra indicate reversibility. We demonstrate these thresholds are stable across models, datasets, and methods.
>
> - Practical guidance, e.g.: i) track mean PCA distance and CKA during unlearning to decide when to stop or adjust LR/request budgets before crossing into irreversible collapse, ii) use drift metrics to tune learning rate and request counts to target reversible or irreversible regimes deliberately, and iii) use Fisher shifts and layer-wise drift to localize updates, preserving retain/unrelated representations while acting on forget-specific directions.
>
> In short, beyond a correlation, we provide: i) a standardized protocol to assess reversibility; ii) layer-wise diagnostics and a scalar summary to set practical parameters; and iii) procedures for monitoring, tuning, and targeting updates. These constitute actionable tools for practitioners aiming to avoid superficial "suppression" and to make progress toward the genuinely useful goal of **irreversible, non‑catastrophic unlearning.**

---

> ### Author Response · Authors · 2025-11-14
> **Response to W3, Q2, W4, W5 and Q3**
>
> **#W3&Q2: Clarifying the Role and Scope of Our Theoretical Analysis**
>
> Thank you for the thoughtful critique. We clarify the role and scope of our theory below.
>
> The theory does more than restate "more perturbation => more dissimilarity." We model unlearning as structured, layer-wise perturbations and formalize how localized, small updates yield first-order effects (hence recoverable), whereas distributed, multi-layer updates accumulate higher-order interactions that drive persistent representational drift. We connect this to our diagnostics via standard perturbation bounds (e.g., Davis–Kahan for PCA similarity, Gram-matrix perturbations for CKA, and Fisher shifts), providing a formal basis for the empirical regime transitions we observe.
>
> We define irreversibility with respect to a fixed, "budgeted" relearning protocol. The theory explains that when accumulated, misaligned perturbations are sufficient to move the model beyond the recoverable region under this protocol, thereby linking representational drift to our operational criterion for irreversible, non-catastrophic forgetting.
>
> Our point is not the trivial last-layer-randomization example; we agree that it can "scramble" outputs while preserving upstream knowledge. Our claim concerns realistic unlearning trajectories: we show runs with similar task-level accuracy can exhibit markedly different representation drift and relearning difficulty. Hence, output-level behavior alone cannot distinguish the four regimes in Table 1; the layer-wise PCA/CKA/Fisher signals do.
>
> In sum, the theoretical section provides a structured account of how multi-layer perturbations translate into recoverable versus non-recoverable drift under a bounded relearning protocol, and it formally underpins the empirical bands of reversibility we document.
>
> **#W4&Q3: Clarifying the Scope and Limitations of Our Experimental Setting**
>
> Our scope is intentional: we aim to diagnose the fundamental reversibility of current unlearning methods using an attack-agnostic, representation-level framework. When unlearning is reversible, forgotten information persists in internal representations and can be recovered through various methods, including optimization-based relearning (ours), prompt / jailbreak attacks [b], or interpretability-driven extraction [c]. These differ in mechanisms but share the same premise.
>
> Our contribution is to characterize when and why this reversibility occurs, via a unified taxonomy and layer-wise diagnostics (PCA similarity/shift, CKA, Fisher), and to identify what would be required for irreversible, non‑catastrophic forgetting. The framework applies independently of the attack surface and explains recovery behavior observed across data sources (forget, retain, unrelated).
>
> We'll add a discussion of prompt-based and interpretability-based recovery in our revision and include an evaluation to illustrate how these probes align with the reversibility patterns.
>
> **#W5&Q3: Justifying the Representativeness of the Evaluated Unlearning Methods**
>
> Exhaustively covering all unlearning methods is infeasible, so we selected methods that represent the dominant design paradigms in LLM unlearning:
>
> - Gradient-ascent family (GA, GA+GD, GA+KL): classical loss-increase approaches that underlie many recent methods [3,4,5].
>
> - Constrained optimization (NPO, NPO+KL): explicitly balancing forget/retain objectives [6].
>
> - Target rewriting (RLabel): canonical label/target corruption strategy [4,7].
>
> These families capture the core mechanisms used in practice.
>
> Pilot experiments with continual unlearning variants such as $O^3$ [8] and ALKN [9] further show that these tailored approaches exhibit the same qualitative reversibility patterns as GA and NPO. This suggests that the regimes we identify arise from structural properties of the unlearning problem itself, rather than from the specifics of any single algorithm. We'll clarify this in our revised version.

---

> ### Author Response · Authors · 2025-11-14
> **Response to W6 and W7**
>
> **#W6: Interpreting the Results in Section 5.3**
>
> We share the goal of disentangling "seemingly irreversible due to weak relearning" from genuinely irreversible, non‑catastrophic unlearning.
>
> We define irreversibility relative to a fixed, stress‑tested relearning budget. In Sect 5.3, we vary the relearning setup along multiple axes: data source (forget/retain/unrelated), budget (unlearning requests and learning rate), while holding retain utility constraints. Once the model crosses into the irreversible, non‑catastrophic regime, recovery fails consistently across these relearning settings, including full forget‑set relearning. Increasing the budget further either leaves recovery unchanged or pushes the model into catastrophic degradation.
>
> These failures coincide with large, misaligned representational drift: substantial PCA rotation/centroid shift and diminished Fisher mass in layers tied to the forget features, while retain geometry remains comparatively stable. This pattern is incompatible with merely weak relearning and is replicated across architectures (see Reviewer HtjF's Tables*4–5).
>
> We agree that an ideal notion would hold under unbounded relearning. Current methods reach a "seemingly irreversible under bounded relearning" state. If requested, we'll expand Sect-5.3 with budget-response curves across different model results to further separate relearning constraints from genuinely irreversible, non‑catastrophic behavior and to inform designs that target this regime (in our revision).
>
> **#W7: Clarifying the Purpose and Value of the Layer-Wise Similarity Plots**
>
> Thanks for the suggestion. The supplementary plots are intended as layer-wise visualizations that substantiate the trends summarized in Tables 2, 3, and 7. Their interpretation is provided in Appendix A.5. They make the representation-level behavior transparent across metrics and architectures.
>
> We'll streamline this section in our revision by reducing redundancy (e.g., retaining representative layers for Fisher Information and PCA/CKA), and add clearer pointers linking each plot group to the corresponding main-text results. If further consolidation is preferred, we can provide a compact summary figure and move detailed plots to an online repository.
>
> [1] Large language models relearn removed concepts. In Findings of ACL 2024.
>
> [2] Eight methods to evaluate robust unlearning in llms. arXiv:2402.16835.
>
> [3] MUSE: machine unlearning six-way evaluation for language models. ICLR 2025.
>
> [4] Machine unlearning of pre-trained large language models. ACL 2024.
>
> [5] TOFU: A task of fictitious unlearning for llms. COLM 2024.
>
> [6] Negative preference optimization: From catastrophic collapse to effective unlearning. arXiv:2404.05868.
>
> [7] Who’s harry potter? approximate unlearning in llms. arXiv:2310.02238.
>
> [8] Adaptive localization of knowledge negation for continual llm unlearning. ICML 2025.
>
> [9] On large language model continual unlearning. ICLR 2025.

---

### Official Review · Reviewer_Ese5 · 2025-10-29

**Soundness:** 3
**Presentation:** 2
**Contribution:** 2
**Rating:** 4
**Confidence:** 4

**Summary:**

This paper studies two aspects of machine unlearning: reversibility and catastrophic forgetting.
Reversibility is probed via relearning, i.e., a one-epoch finetune on the forget set.
The authors show that, in many cases, this simple adjustment of model weights by relearning restores much of the original model’s performance.

From this, they infer that the knowledge was not effectively removed and can be readily recovered. They further support this by analyzing the intermediate representation space of LLMs and measuring how the unlearned-then-relearned model deviates from the original.

Overall, the paper finds that relearning often recovers performance, presenting this as a failure mode of unlearning and evidence that the underlying knowledge persists.

**Strengths:**

The idea of studying how easily unlearned knowledge can be recovered after unlearning is quite interesting. In particular, applying relearning and then evaluating the model’s recovery is a valuable direction that deserves further exploration.

**Weaknesses:**

I don’t find the results of this paper particularly surprising. A single step of finetuning on the forget set can naturally bring back the forgotten knowledge. I don’t quite see why the authors expected this not to work.
After all, with more aggressive settings (e.g., two or three additional epochs), one could almost certainly recover the utility on the forget set. Restoring performance through one epoch of finetuning is not unexpected.

In general, unlearning methods that are truly “irreversible” often achieve this by severely degrading the model, seen as a drop in accuracy on the retain set and overall utility.
A more interesting direction, in my view, would be to study the sample efficiency of relearning: can we recover performance using only a few samples or perhaps by providing them as in-context examples instead of full retraining?

Also, I recenlty found a paper on knowledge recovey of machine unlearning [1], I guess this also worth being discussed in this paper.
[1] Rezaei, Keivan, et al. "RESTOR: Knowledge Recovery in Machine Unlearning." arXiv preprint arXiv:2411.00204 (2024).

**Questions:**

They are discussed in the weakness section.

---

> ### Author Response · Authors · 2025-11-14
> **Response to W1**
>
> We thank the reviewer for the constructive comments and for recognizing our strengths, including **the relevance of studying knowledge recovery through relearning**.
>
> **#W1: Clarifying the Role of Relearning and the Interpretation of “Irreversible” Unlearning**
>
> Thank you for the thoughtful feedback and for bringing up RESTOR; we'll discuss it in our revision and relate it to our findings.
>
> **Scope and contribution.** Our aim is not to show that fine-tuning can recover performance. This is well known and consistent with prior work [1,2]. To our knowledge, this is the **first study** to systematically diagnose unlearning with four complementary representation-level tools (PCA similarity/shift, CKA, Fisher information), establishing a taxonomy of four regimes (Table 1) and demonstrating that task-level metrics (like accuracy) are insufficient for assessing unlearning effectiveness. Across single and continual settings, existing methods overwhelmingly fall into **reversible** regimes. The more challenging, interesting, and practically useful target is **irreversible, non-catastrophic** forgetting; we are not aware of any method that reliably achieves it. Simply "severely degrading the model" belongs to the "catastrophic" category in our taxonomy and is not the objective.
>
> **Role of relearning.** In our framework, relearning is just an empirical probe (or "attack") to reveal reversibility, not an end goal. We show that recovery can be highly sample-efficient: even small fractions (e.g., 10%) of the forget set, retain-set-only relearning, or unrelated-data relearning can restore behavior under mild settings, highlighting how easily suppressed knowledge resurfaces and why task-level recovery alone is misleading.
>
> **Future directions.** We agree that sample efficiency and in-context recovery are important; we'll add a compact analysis of recovery curves and an in-context probe. We'll also integrate RESTOR into Related Work, clarifying how its recovery results align with our reversibility diagnosis and taxonomy, and how they underline the open challenge of achieving irreversible, non-catastrophic unlearning.
>
> [1] Large language models relearn removed concepts. In Findings of ACL 2024.
>
> [2] Eight methods to evaluate robust unlearning in llms. arXiv:2402.16835.

---

> > ### Comment · Reviewer_Ese5 · 2025-11-24
> >
> > Thanks for the rebuttal.
> >
> > I still think a more comprehensive study of different types of relearning, sample-efficieny, or in-context recovery can strengthen the contribution of this paper.
> >
> > I'd be happy to raise my score if I see experimental results regarding that, but I think the paper could really be strong if augmented with those future directions that the author mentioned.
> >
> > I want to keep my score as is for now.

---

> > > ### Author Response · Authors · 2025-11-25
> > > **Response [1/2]**
> > >
> > > We sincerely appreciate your continued engagement and the specific suggestion to expand our experimental scope. We agree that a more comprehensive study strengthens the paper’s contribution. In response, we have conducted the requested experiments regarding **alternative recovery mechanisms** and **sample efficiency**.
> > >
> > > 1. **Different types of relearning:**
> > >
> > > Beyond standard relearning, we evaluated the unlearned model (Yi-6B, GA-based setup) against **four** additional recovery strategies: quantization attack [2], prompt attack (Reviewer n9vs) [1], jailbreak attack [3], and in-context recovery (as suggested).
> > >
> > > For **quantization**, we directly applied Int4 quantization. For the other three (which do not modify parameters), we adapted inputs to fit our PCA analysis:
> > >
> > > - **prompt attack:** we used an input-rephrasing variant that paraphrases the original text.
> > > - **jailbreak attack:** we prepended the fixed prefix used in [3] to each original input.
> > > - **in-context recovery:** before PCA analysis, we provided five demonstrations from the forget set and then evaluated the model on the original inputs.
> > >
> > > As shown in the new results (Table*1), **none of these methods successfully restore the forgotten knowledge**. Once the model enters the regime of reversible catastrophic forgetting, methods that do not explicitly update parameters (prompting, jailbreaking, in-context) or only apply minor perturbations (quantization) fail to recover lost representations. This confirms that explicit relearning is necessary to reverse this specific forgetting state.
> > >
> > >
> > > 2. **Sample efficiency (and data types/sources)**:
> > >
> > > To address your concern on sample efficiency, we expanded our GA-based relearning experiments ($lr = 6\times10^{-6}$, $N=100$) using three distinct data sources: the **forget set**, **retain set**, and **unrelated data** (see also lines 157--161 for their description). We evaluated each at 10%, 30%, 60%, and 100% of the original forget-set size.
> > >
> > > We observe a distinct hierarchy in recovery efficiency (Table*2):
> > >
> > > - **Forget-set**: Achieves the fastest and strongest recovery, with PCA distances approaching the original model even at moderate sample sizes.
> > > - **Retain-set** and **unrelated data**: While recovery is possible, these sources are significantly less sample-efficient and exhibit slower improvement.
> > >
> > > In short, these additional experiments demonstrate that while explicit relearning (specifically with the forget set) is highly efficient, alternative "lightweight" recovery mechanisms fail once catastrophic forgetting occurs. We'll incorporate them in our revision to be uploaded soon. (As suggested by Reviewer HtjF, we have included cross-model evidence in the revised PDF to further support the generality of our findings.)
> > >
> > > We hope these new findings address your concerns and merit an improved score.
> > >
> > > Table*1
> > > | Setting(Yi-6B GA, lr=6e-6, N=100)                         | F.Acc | mean PCA distance (forget set) |
> > > |---------------------------------|-------|---------------------------------|
> > > | **Original model**              | 78.90 | 0.00                            |
> > > | **Unlearned model**             | 0.00  | 31.66                           |
> > > | **Quantization**                |0.00   |  32.21                       |
> > > | **In-context (num_demos = 5)**  | 0.01  | 30.83                           |
> > > | **Prompt attack**               | 0.03    |        29.14                 |
> > > | **Jailbreaking**                | 0.03  | 30.04                           |

---

> > > ### Author Response · Authors · 2025-11-25
> > > **Response [2/2]**
> > >
> > > Table*2
> > > | Setting(Yi-6B GA, lr=6e-6, N=100)                         | F.Acc | mean PCA distance (forget set) |
> > > |---------------------------------|-------|--------------------------------|
> > > | **Original model**              | 78.90 | 0.00                           |
> > > | **Unlearned model**             | 0.00  | 31.66                          |
> > > |                                 |       |                                |
> > > | **Relearned by forget set**     |       |                                |
> > > | 10%                             | 67.28 | 8.49                           |
> > > | 30%                             | 75.77 | 6.42                           |
> > > | 60%                             | 77.13 | 4.31                           |
> > > | 100%                            | 79.20 | 2.16                           |
> > > |                                 |       |                                |
> > > | **Relearned by retain set**     |       |                                |
> > > | 10%                             | 0.05  | 30.57                          |
> > > | 30%                             | 11.24 | 25.48                          |
> > > | 60%                             | 45.24 | 14.69                          |
> > > | 100%                            | 75.86 | 7.51                           |
> > > |                                 |       |                                |
> > > | **Relearned by unrelated data** |       |                                |
> > > | 10%                             | 0.02  | 31.02                          |
> > > | 30%                             | 6.48  | 27.74                          |
> > > | 60%                             | 38.83 | 17.51                          |
> > > | 100%                            | 65.66 | 9.14                           |
> > >
> > > [1] Can sensitive information be deleted from llms? objectives for defending against extraction attacks. ICLR 2024.
> > >
> > > [2] Catastrophic Failure of LLM Unlearning via Quantization. ICLR 2025.
> > >
> > > [3] Jailbreaking ChatGPT via Prompt Engineering: An Empirical Study. arXiv:2305.13860

---

> > > ### Author Response · Authors · 2025-11-26
> > >
> > > We have uploaded the latest revised manuscript.
> > >
> > > The requested experiments are supplemented in Appendices A.4.1 and A.4.2, with related descriptions in lines 80--87 and 154--161. RESTOR [1] is now discussed in lines 802--804.
> > >
> > > Please let us know if there are any further comments you would like us to address.
> > >
> > > [1] RESTOR: Knowledge Recovery in Machine Unlearning. Trans. Mach. Learn. Res., 2025.

---

### Official Review · Reviewer_HtjF · 2025-10-30

**Soundness:** 3
**Presentation:** 3
**Contribution:** 2
**Rating:** 6
**Confidence:** 3

**Summary:**

This paper highlights the drawback of relying solely on task-level metrics (e.g., accuracy, perplexity) for evaluating unlearning in LLMs, since these metrics cannot distinguish genuine erasure from superficial forgetting. To bridge this evaluation gap, the paper introduces a **representation-level analysis framework** to measure representational drift and categorize unlearning behavior into four regimes. The study concludes that achieving the ideal state—**irreversible and non-catastrophic forgetting**—is extremely challenging, and further provides a method combination that achieves a *seemingly* irreversible, non-catastrophic form of forgetting.

**Strengths:**

- The paper is well-written and easy to follow.
- It clearly identifies the limitations of current task-level evaluations and proposes a **representation-level toolkit** that goes beyond surface metrics.
- Provides clear definitions and a systematic taxonomy of forgetting regimes.

**Weaknesses:**

- Table 2 demonstrates the weakness of task-level metrics, but it would be stronger to include results on the **Qwen2.5-7B** model to further consolidate this finding.
- It remains unclear whether the same observations hold for **smaller (3B) or other model families (Llama)**.
- The framework measures representational drift but does not formally assess **privacy leakage**; the notion of “irreversible forgetting” is still heuristic.
- The proposed solution is interesting, but **cross-model validation** would strengthen its generality.

**Questions:**

- Does the proposed framework also generalize to **LLaMA** or **Qwen3** models?
- Could the **mean PCA distance** be correlated with formal privacy metrics such as **MIA AUC** in a consistent way?
- In Tables 2 and 3, how relearning is conducted?

---

> ### Author Response · Authors · 2025-11-14
> **Response to W1, W2, W3 and Q2**
>
> We thank the reviewer for the constructive comments and for recognizing our strengths, including **the clear presentation, the identification of task-level limitations, and the introduction of a systematic representation-level toolkit.**
>
> **#W1: Results on Qwen2.5-7B to further validate limitations of task-level metrics**
>
> We conducted additional single-unlearning experiments on Qwen2.5-7B. Due to time constraints, we evaluated the GA method at two learning rates. As shown in Table*1, both settings achieve full recovery in task-level accuracy, consistent with the **reversible, non-catastrophic forgetting regime.** These findings further suggest that task-level metrics alone are insufficient to assess the true reversibility or stability of unlearning. Given the rebuttal policy, we will include the corresponding PCA-shift visualizations in our revision.
>
> Table*1
> | Qwen2.5-7B (GA)       | MATH | GSM8K |
> |-----------------------|-------|-------|
> | Original model        |  9.00 | 80.10 |
> | 3×10⁻⁶ (unlearn)      |  6.24 | 73.28 |
> | 3×10⁻⁶ (relearn)      |  8.97 | 78.29 |
> | 6×10⁻⁶ (unlearn)      |  1.12 |  30.21 |
> | 6×10⁻⁶ (relearn)      |  8.62 |  77.63 |
>
>
> **#W2: Observations hold for smaller models (3B) and or other families (LLaMA)?**
>
> We further conducted continual-unlearning experiments (N=100) on Llama-3-8B and Qwen2.5-3B using GA at two learning rates, following the setup in Table 3. As shown in Tables*2 and *3, both models exhibit the two regimes observed in Yi-6B: with a low learning rate, **reversible catastrophic forgetting**, where task-level accuracy appears to fully recover; with a high learning rate, **irreversible catastrophic forgetting**, where both forget and retain performance collapse.
>
> These findings indicate that the behaviors in Table 3 are not specific to Yi-6B and generalize across architectures and parameter scales, underscoring that task-level metrics alone do not reliably capture the reversibility or stability of unlearning. Likewise, we will include the corresponding PCA-shift visualizations in our revision.
>
> Table*2
>  |LLaMA-3-8B (GA) | F.Acc | R.Acc |
> |----------------------|--------|--------|
> | Original model       | 76.41  | 63.50  |
> | 6×10⁻⁶ (unlearn)     | 0.38   | 0.48   |
> | 6×10⁻⁶ (relearn)     | 76.49  | 63.21  |
> | 5×10⁻⁵ (unlearn)     | 0.00   | 0.00   |
> | 5×10⁻⁵ (relearn)     | 0.02   | 0.04   |
>
> Table*3
> |Qwen2.5-3B (GA)              | F.Acc | R.Acc |
> |----------------------|--------|--------|
> | Original model       | 76.37  | 61.39  |
> | 6×10⁻⁶ (unlearn)     | 1.45   | 2.56   |
> | 6×10⁻⁶ (relearn)     | 79.61  | 61.45  |
> | 5×10⁻⁵ (unlearn)     | 0.01   | 0.01   |
> | 5×10⁻⁵ (relearn)     | 3.58   | 4.27   |
>
> **#W3&Q2: The framework measures representational drift but does not formally assess privacy leakage**
>
> **Scope and contribution.** Our work introduces a representation-level diagnostic toolkit to analyze the reversibility and catastrophicity of unlearning. We show that task-level metrics alone are often misleading and cannot distinguish reversible from irreversible forgetting. Our objective is not to provide formal privacy guarantees; rather, we offer operational tools for diagnosing when unlearning is merely suppressive vs. structurally altering representations.
>
> **On formal privacy.** Establishing differential-privacy-style or certified removal guarantees for LLM unlearning is non-trivial: modern LLM training and unlearning are non-convex and lack strong convexity/smoothness assumptions that underlie most DP or certified-removal analyses. Consequently, current practice relies on empirical proxies, including MIAs, which do not constitute formal guarantees.
>
> **Irreversibility is operational, not heuristic.** We define irreversibility via a controlled relearning protocol: unlearning is irreversible when the model fails to recover its original representations and performance under a fixed, "budgeted" relearning process. This is a protocol-defined criterion that complements (not replaces) formal privacy analyses.
>
> **Mean PCA distance vs. MIA AUC:**
> - Mean PCA distance: representation drift across layers after unlearning/relearning.
> - MIA AUC: attack separability between members and non-members, dependent on attack design and model calibration.
>
> Correlation can appear within a fixed regime (same model, layer, PCA rank) when unlearning truly reduces memorization by moving forget representations toward non-member manifolds; in such cases, larger drift may coincide with lower MIA AUC. However, our results (e.g., Tables 2--3) show the relationship is not consistent across methods and hyperparameters. MIA outcomes are sensitive to distributional contrasts between forget and retain sets as well as to calibration shifts. We also observe regimes where MIA AUC rebounds despite persistent representational collapse, and vice versa. Hence, a universal, monotone correlation should not be expected.

---

> ### Author Response · Authors · 2025-11-14
> **Response to W4, Q1 and Q3**
>
> **# W4: Cross-Model Evidence Supporting the Generality of Our Findings**
>
> To assess generality, we replicated the Table-5 setup with GA+GD+WAGLE under continual unlearning, using an unrelated dataset for relearning due to time constraints.
>
> Supplementary Table*4 and *5 closely match Table 5: we observe **seemingly irreversible, non-catastrophic forgetting.** The forget set exhibits large, unrecoverable PCA distances under the bounded relearning protocol, while the retain set shows modest, partially recoverable degradation. This underscores that targeted, permanent unlearning without collateral damage remains an open challenge.
>
> We also note model-family differences in hyperparameter sensitivity, especially learning rate, to reach comparable regimes. This model-specific sensitivity likely shapes the boundary between reversible and irreversible unlearning and may inform future efforts toward achieving **irreversible, non-catastrophic forgetting**.
>
> Table*4
> | Qwen3-8B-Base (relearned by unrelated data) | F.Acc | R.Acc | Mean PCA distance (forget set) | Mean PCA distance (retain set) |
> |---------------------------------------------|-------|-------|---------------------------------|---------------------------------|
> | Original model                              | 78.28 | 62.96 | 0.00                            | 0.00                            |
> | unlearn (LR = 5×10⁻⁶, N = 50)               | 48.52 | 56.47 | 8.49                            | 5.98                            |
> | relearn (unrelated data, N = 50)            | 53.21 | 59.16 | 6.57                            | 4.32                            |
>
> Table*5
> | Llama-3-8B (relearned by unrelated data)    | F.Acc | R.Acc | Mean PCA distance (forget set) | Mean PCA distance (retain set) |
> |---------------------------------------------|-------|-------|---------------------------------|---------------------------------|
> | Original model                              | 76.41 | 63.50 | 0.00                            | 0.00                            |
> | unlearn (LR = 3×10⁻⁶, N = 50)               | 42.59 | 53.47 | 14.29                           | 7.12                            |
> | relearn (unrelated data, N = 50)            | 49.78 | 56.24 | 11.47                           | 6.21                            |
>
>
> **#Q1: Generalization Across other models**
>
> Besides the supplemented results on Qwen2.5-7B,  Qwen2.5-3B, and Llama-3-8B, we evaluated our framework on Qwen3-8B-Base using GA with two learning rates. As shown in Tables*1–*6, this model exhibits the same three forgetting regimes and the same reversibility patterns as in our other evaluations. These findings further indicate that our framework generalizes across architectures and model families.
>
>
> Table*6
> | Qwen3-8B-Base (GA)     | F.Acc | R.Acc |
> |------------------------|-------|-------|
> | Original model         | 78.28 | 62.96 |
> | 6×10⁻⁶ (unlearn)       | 0.45  | 0.21  |
> | 6×10⁻⁶ (relearn)       | 79.72 | 62.66 |
> | 5×10⁻⁵ (unlearn)       | 0.02  | 0.02  |
> | 5×10⁻⁵ (relearn)       | 0.03  | 0.03  |
>
> **#Q3: Relearning Procedure in Tables 2 and 3**
>
> As discussed in Sect-3.1 (Experiment Setup), in the single-unlearning setting, we relearn by fine-tuning once on the entire forget set, using the same hyperparameters as in unlearning.  For Table 3, the setting is continual unlearning: the model processes a sequence of requests, and the relearning phase uses the cumulative forget set. Aside from this difference, all training hyperparameters are identical to those used during unlearning.
>
> We will upload the revised manuscript after finalizing all responses. Please let us know if any further clarification would be helpful. Thank you again for the insightful and constructive feedback.

---

### Author Response · Authors · 2025-12-01
**Summary of Contribution, Resolved Reviewer Concerns, and Revision**

Dear AC (and Reviewers),


We sincerely thank all reviewers for their constructive comments. To facilitate your decision-making and meta-review, we briefly recap the current status below.

We first reiterate our key contributions, several of which are also recognized by reviewers:

1. **First systematic, representation-level analysis** of LLM unlearning reversibility, highlighting the value of studying how easily forgotten knowledge can be recovered via relearning (Reviewer $\textcolor{Maroon}{Ese5}$).

2. **Identified key limitations of task-level evaluations** and introduced a representation-level toolkit that captures underlying drift beyond surface metrics (Reviewer $\textcolor{Maroon}{HtjF}$).

3. **Provided clear definitions and a systematic taxonomy** of practical forgetting regimes (Reviewer $\textcolor{Maroon}{HtjF}$; Reviewer $\textcolor{Maroon}{n9vs}$).

4. **Demonstrated that existing methods fail to achieve irreversible and non-catastrophic unlearning** (Reviewer $\textcolor{Maroon}{n9vs}$).

5. **Presented a concrete example of successful irreversible, non-catastrophic unlearning**, showing that this goal is attainable under certain conditions (Reviewer $\textcolor{Maroon}{n9vs}$).

6. **Further comprehensively analysed multiple relearning mechanisms and the sample efficiency of relearning with different data types**, revealing how each contributes to recovering forgotten knowledge.

During the rebuttal, we received a response from Reviewer $\textcolor{Maroon}{Ese5}$, whose initial rating is 4 but **is willing to increase the score**, indicating that the additions on relearning types and sample efficiency would strengthen the evaluation; these have been now been incorporated.

For Reviewer $\textcolor{Maroon}{HtjF}$ (Rating: 6), whose primary concern was the generality of our framework and findings, we have added experiments that directly address this point.

For Reviewer $\textcolor{Maroon}{n9vs}$ (Rating: 2), the concerns focused on interpretation, presentation, and experimental setting; we have clarified the underlying concepts and added supporting experiments where appropriate. While no further responses have been received, we believe the main concerns have been substantively addressed, including:

- Validating the limitations of task-level metrics with Qwen2.5-7B results(Reviewer $\textcolor{Maroon}{HtjF}$ #W1)

- Demonstrating reversibility across smaller and cross-family models: Qwen2.5-3B, Llama-3-8B, and Qwen3-8B-Base (Reviewer $\textcolor{Maroon}{HtjF}$ #W2 & #Q1)

- Providing cross-model evidence on seemingly irreversible yet non-catastrophic forgetting (Llama-3-8B, Qwen3-8B-Base) (Reviewer $\textcolor{Maroon}{HtjF}$ #W4)

- Clarifying that our framework measures representational drift (not privacy leakage) and that irreversibility is operationally defined (Reviewer $\textcolor{Maroon}{HtjF}$ #W3 & Q2)

- Expanding the analysis of relearning mechanisms and sample efficiency (Reviewer $\textcolor{Maroon}{Ese5}$ #W1; Reviewer $\textcolor{Maroon}{n9vs}$ #W4)

- Strengthening the originality and contribution of our framework and analysis (Reviewer $\textcolor{Maroon}{n9vs}$ #W1 & Q1)

- Elaborating practical insights and actionable takeaways from the representation analysis (Reviewer $\textcolor{Maroon}{n9vs}$ #W2 & Q2)

All these revisions have been incorporated into the updated manuscript. We thank the reviewers again for their thoughtful feedback, which has helped improve the clarity and depth of the work. We believe the revised manuscript makes a meaningful contribution to LLM unlearning research.

Thank you for your consideration.

Best regards,
Authors

---

### Meta-Review · Area_Chair_oUNb · 2025-12-25

**Summary:**

This paper conducts a representation-level analysis of unlearning behavior. It proposed a unified analysis framework and a taxonomy of forgetting regimes, exposing the limitations of task-level metrics and providing a more reliable diagnostic toolkit. The theoretical analysis (perturbation model) and experimental validation explain why some forgetting is reversible while some is not.

**Reviewer Concerns:**

For reviewer HtjF, most concerns have been addressed. The experiments are comprehensive, and the explanations are reasonable. For reviewer Ese5, some concerns have been partially addressed. While the experiments are enriched, the responses lack thoroughness. As for reviewer n9vs, I believe the concerns have also been partially resolved. The experiments and explanations have been strengthened; however, lingering concerns about originality remain.

**Reviewer Scores:**

1. HtjF: 6 (marginally above the acceptance threshold) - Likely to maintain;
2. Ese5: 4 (marginally below the acceptance threshold) - Likely to maintain or increase;
3. n9vs: 2 (reject) - Likely to maintain or slightly increase.

The primary concerns are its originality and the comprehensiveness of the investigation.

Regarding originality, I agree with a prior comment that the potential recoverability of “forgotten” knowledge, when internal representations stay near their origin, is already recognized in the literature, even amidst a task-performance gap. Thus, the finding that more recoverable models show smaller representation-level distances is, to some extent, expected. This core originality issue remains insufficiently addressed in the authors’ rebuttal.

Regarding analytical depth, the paper currently relies heavily on qualitative observations and lacks rigorous quantitative evaluation. A critical next step would be to formally model the relationship between representation-level distance and task performance, rather than merely reporting their correlation. Furthermore, the proposed actionable guidelines (e.g., “Drift metrics help tune learning-rate schedules and request counts”) would benefit from experimental validation to substantiate their practical utility.

In summary, while the work applies a representation-level lens to unlearning, it would require substantial strengthening in both novelty and analytical thoroughness to make a compelling contribution.

---

### Decision · Program_Chairs · 2026-01-26

Reject